# Iron control of erythroid microtubule cytoskeleton as a potential target in treatment of iron-restricted anemia

Adam N. Goldfarb [1✉], Katie C. Freeman[1], Ranjit K. Sahu[1], Kamaleldin E. Elagib[1], Maja Holy[1], Abhinav Arneja[1], Renata Polanowska-Grabowska[2], Alejandro A. Gru [1], Zollie White III[1], Shadi Khalil[3], Michael J. Kerins[4], Aikseng Ooi[4], Norbert Leitinger[2], Chance John Luckey[1] & Lorrie L. Delehanty[1]

Anemias of chronic disease and inflammation (ACDI) result from restricted iron delivery to erythroid progenitors. The current studies reveal an organellar response in erythroid iron restriction consisting of disassembly of the microtubule cytoskeleton and associated Golgi disruption. Isocitrate supplementation, known to abrogate the erythroid iron restriction response, induces reassembly of microtubules and Golgi in iron deprived progenitors. Ferritin, based on proteomic profiles, regulation by iron and isocitrate, and putative interaction with microtubules, is assessed as a candidate mediator. Knockdown of ferritin heavy chain (FTH1) in iron replete progenitors induces microtubule collapse and erythropoietic blockade; conversely, enforced ferritin expression rescues erythroid differentiation under conditions of iron restriction. Fumarate, a known ferritin inducer, synergizes with isocitrate in reversing molecular and cellular defects of iron restriction and in oral remediation of murine anemia. These findings identify a cytoskeletal component of erythroid iron restriction and demonstrate potential for its therapeutic targeting in ACDI.

[1] Department of Pathology, University of Virginia School of Medicine, Charlottesville, VA, USA. [2] Department of Pharmacology, University of Virginia School of Medicine, Charlottesville, VA, USA. [3] Department of Dermatology, University of California, San Diego School of Medicine, San Diego, CA, USA. [4] Department of Pharmacology and Toxicology, College of Pharmacy, University of Arizona, Tucson, CA, USA. ✉email: ang3x@virginia.edu

The selective repression of marrow red cell production in response to declining iron availability, the erythroid iron restriction response, underlies the two most common human anemias: anemia of chronic disease and inflammation (ACDI) and iron deficiency anemia (IDA)[1–3]. These anemias confer major domestic and global disease burdens[4,5]. Although many novel treatment strategies have been developed for ACDI[2,6,7], conventional therapies remain the standard of care despite concerns regarding their safety, cost, efficacy, and convenience[6,8,9]. An early molecular step in the erythroid iron restriction response consists of inactivation of the iron-dependent aconitase enzymes[10,11]. Supplying cells with the downstream product isocitrate abrogates the erythropoietic blockade caused by iron restriction[11–13]. In several rodent models, isocitrate treatment demonstrates potential to ameliorate anemia[12,14], but high doses via intraperitoneal injection are required and yield transient benefits[14].

A key element in the erythroid iron restriction response comprises progenitor resistance to erythropoietin (Epo), clinically reflected in poor patient responses to Epo treatment[15,16]. Mechanistically, the iron deprivation decreases cell surface delivery of the Epo receptor (EpoR) and associated factors (Scribble and TfR2), in a manner reversible by isocitrate[13]. Restoring Epo responsiveness to iron-restricted erythroid progenitors breaks a vicious cycle of erythroid repression in ACDI. Specifically, Epo induction of erythroid ERFE secretion counteracts the hepcidin overproduction, a key step in ACDI pathogenesis, and restores iron bioavailability without causing iron overload[17]. By contrast, treatments using intravenous (IV) iron administration or hepcidin neutralization bypass the ERFE–hepcidin circuit and drive iron overload while promoting erythropoiesis. Thus, identifying clinically feasible strategies to reverse the erythroid iron restriction response will provide safe alternatives for ACDI therapy.

In current studies assessing factors that might affect EpoR vesicular trafficking, we found that iron restriction caused erythroid Golgi and microtubule disruption, both in cultured progenitors and in clinical specimens. Isocitrate supplementation did not prevent cytoskeletal collapse, but induced gradual microtubule recovery. To explore the basis for iron control of erythroid microtubules, we interrogated the human erythroblastic proteome[18,19] for microtubule-associated proteins (MAPs) and found predominance of the destabilizer Stathmin 1 (STMN1) with virtual absence of stabilizing MAPs. However, erythroid cells abundantly expressed a candidate noncanonical stabilizing MAP, ferritin heavy chain (FTH1), previously suggested to have microtubule bundling activity[20]. In erythroid progenitors, FTH1 strongly declined with iron restriction and underwent rescue with isocitrate supplementation. FTH1 knockdown in iron-replete progenitors recapitulated the phenotypic features of iron restriction, while enforced expression in iron-restricted progenitors rescued differentiation. Implication of FTH1 in coupling iron and erythropoiesis prompted evaluation of fumarate, a known *FTH1* transcriptional inducer[21,22], for potentiation of isocitrate bioactivity. In iron-restricted erythroid progenitors, fumarate enhanced isocitrate rescues of FTH1 expression, microtubule assembly, and differentiation. In a murine model of ACDI, fumarate also synergized with isocitrate in anemia reversal, enabling development of an effective oral treatment regimen.

Here, we demonstrate a lineage-selective nutrient response pathway in which iron availability dictates cytoskeletal integrity through the action of ferritin on microtubules. This pathway is associated with the erythroid iron restriction response that underlies frequent clinical cases of anemia, such as anemias associated with chronic kidney disease. We further demonstrate feasibility of therapeutic targeting through the action of orally bioavailable metabolites on this pathway.

## Results

### The erythroid iron restriction pathway affects microtubule-associated organelle structures

In an attempt to elucidate vesicular trafficking alterations previously described with erythroid iron restriction[13], we evaluated organelle morphology by immunofluorescence microscopy (IF). The Golgi apparatus, detected with anti-Golgin97, possessed a discrete, juxta-nuclear structure in primary human progenitors cultured in iron-replete erythroid medium (transferrin saturation, TSAT, of 100%; Fig. 1a and Supplementary Fig. 1a). By contrast, iron restriction (TSAT of 10%) caused Golgi disruption in the majority (>80%) of cells (Fig. 1a). This disruption appeared, by Airyscan superresolution and electron microscopy, to be associated with organelle fragmentation and collapse (Supplementary Fig. 1a, b). The response was specific to the erythroid lineage, as iron restriction had no effect on granulocytic Golgi (Supplementary Fig. 1c). Isocitrate supplementation has previously been shown to rescue erythroid differentiation and viability under conditions of iron restriction[11,13]. In line with those findings, significant Golgi restoration occurred with isocitrate treatment, even when provided in a delayed manner (Fig. 1a). IF for additional Golgi markers, Giantin, and GM130, yielded similar results, although cis-Golgi structures (marked by GM130) appeared to be less perturbed by iron restriction (Supplementary Fig. 2a, b).

Immunohistochemistry on intact marrow samples addressed in vivo effects of iron restriction. For this approach, Golgi detection required staining of human tissue with anti-Giantin; effective antibodies could not be found for murine marrows. As marrow samples are generally not obtained in IDA cases, comparison groups consisted of ACDI versus non-anemic control patients. In 9/10 ACDI samples, the majority (>50%) of erythroid cells lacked discrete Golgi, while only 1/5 controls displayed this loss (Fig. 1b).

To determine whether iron restriction affects organelles with which Golgi interact[23], erythroid progenitors next underwent IF for centrosomal markers. Co-staining for Pericentrin (PCNT) and Golgin97 confirmed organelle juxtaposition, with Golgi abutting or surrounding centrosomes in iron-replete erythroid progenitors (Fig. 1c). As with Golgi, iron restriction induced centrosomal disruption, and isocitrate reversed this effect (Fig. 1c). As PCNT marks the pericentriolar material (PCM) surrounding the centriole cores, we also conducted IF for CEP135, a centriole component, which showed no clear changes with iron restriction (Supplementary Fig. 2c). A common element involved in maintaining both Golgi and PCM integrity consists of microtubules. In actively cycling cells, microtubules are necessary for post-mitotic Golgi reassembly[24,25] and for centrosomal delivery of PCNT[26,27]. IF for β-tubulin in iron-replete erythroblasts demonstrated a cage-like microtubule cytoskeleton, as described by Koury et al.[28]; this structure underwent dissolution with iron restriction and rescue with isocitrate treatment (Fig. 1d). While some of the iron-restricted erythroblasts retained small tubulin aggregates that appeared to be centriole associated, >90% of the iron-deprived progenitors contained no residual tubular structures (Fig. 1d).

### Dynamic responsiveness of the microtubule cytoskeleton to iron restriction and isocitrate

The extensive disruption of the microtubule cytoskeleton reflected in Fig. 1d suggested a potential role in mediating the erythroid iron restriction response. As with the Golgi changes, the microtubule response showed evidence of lineage selectivity. Granulocytic progenitors at baseline possessed

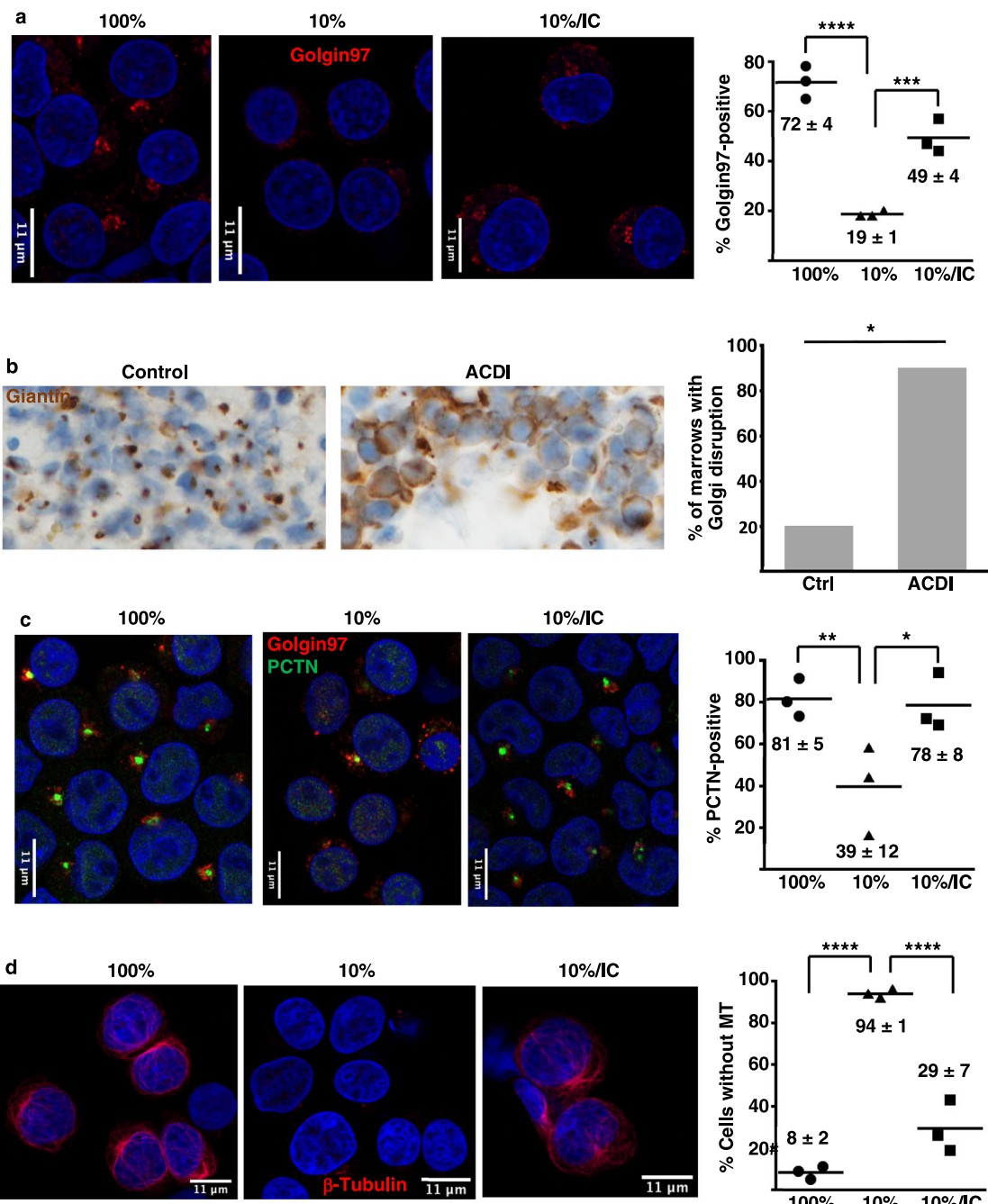

**Fig. 1 Organelle perturbations associated with erythroid iron restriction. a** Documentation of Golgi disruption by immunofluorescence on human CD34+ progenitors cultured 3 days in iron-replete (100% TSAT) or deficient (10% TSAT) erythroid medium ± isocitrate rescue (IC). Red: Golgin97; blue: DAPI. Graph: % cells with detectable Golgi. Numbers indicate mean ± SEM; $n = 3$ biologically independent experiments; ***, ****$P = 0.002$, 0.0001, one-way ANOVA with Tukey post hoc test. **b** Golgi assessment in human marrow samples by immunoperoxidase on clot sections from non-anemic controls and patients with anemia of chronic disease and inflammation (ACDI). 200× magnification. Brown: Giantin; blue: hematoxylin. Graph: % of marrow samples with Golgi disruption in >50% erythroblasts. $n = 5$ for control and 10 for ACDI patient samples; *$P = 0.017$, Fisher's exact two-tailed test. **c** Effects on pericentriolar material (Pericentrin: PCTN) and adjacent Golgi documented by immunofluorescence on cells as in **a**. Red: Golgin97; green: Pericentrin; blue: DAPI Graph: % cells with PCTN+ centrosomes. Numbers indicate mean ± SEM; $n = 3$ independent experiments; *, **$P = 0.049$, 0.037, one-way ANOVA with Tukey post hoc test. **d** Microtubule alterations demonstrated by immunofluorescence on cells as in **a**. Red: β-tubulin; blue: DAPI. Graph: % cells with undetectable microtubule structures (MT). Numbers indicate mean ± SEM; $n = 3$ independent experiments; ****$P = 0.0001$, one-way ANOVA with Tukey post hoc test. See also Supplementary Figs. 1 and 2. Source data are provided as a Source data file.

microtubules that were relatively sparse, short, and disconnected, consistent with structures previously described in mature human neutrophils[29]. We observed no significant changes in granulocytic microtubules in response to iron deprivation (Supplementary Fig. 2d).

Time course analysis of the erythroid response revealed attainment of maximal microtubule disruption within 18 h of initiating iron restriction (Fig. 2a), prior to the reported onset of clear changes in growth, viability, and differentiation[11]. Unexpectedly, isocitrate treatment did not prevent the initial

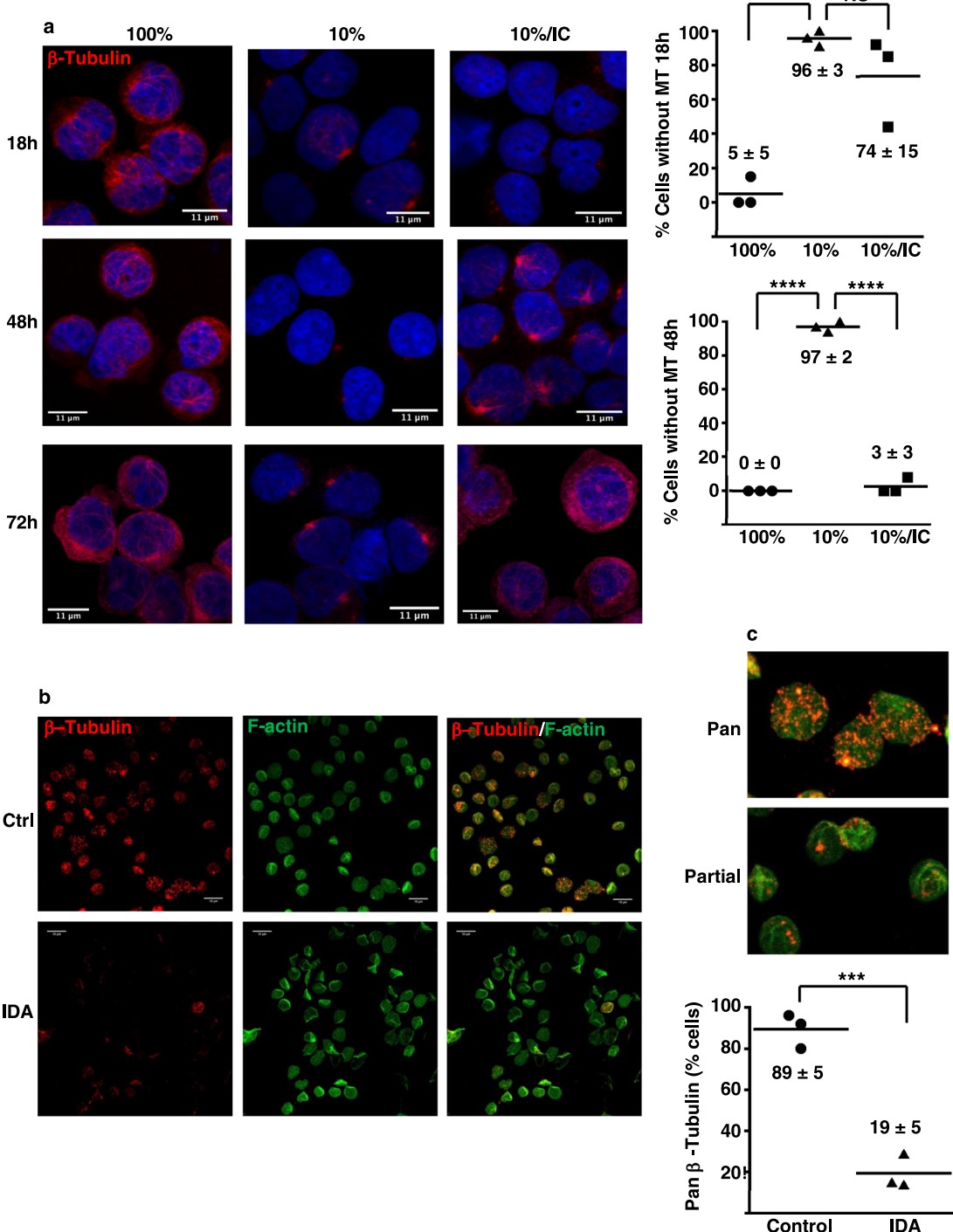

**Fig. 2 Microtubule disruption associated with erythroid iron restriction. a** Microtubule alterations demonstrated by immunofluorescence on human CD34+ progenitors cultured 1–3 days in iron-replete (100% TSAT) or deficient (10% TSAT) erythroid medium ± isocitrate rescue (IC). Red: β-tubulin; blue: DAPI. Graphs: % cells with undetectable microtubule structures (MT) on days 1 and 2. Numbers indicate mean ± SEM; $n = 3$ biologically independent experiments; ***, ****$P = 0.001$, $0.0001$, one-way ANOVA with Tukey post hoc test. **b** Alterations in human erythrocyte microtubule remnants demonstrated by fluorescence staining of peripheral blood samples from an iron-replete non-anemic control (Ctrl) and from a patient with iron deficiency anemia (IDA). White line, 10 μm. Red: β-tubulin; green: filamentous (F) actin. Representative results from three independent experiments. **c** Magnification of images from **b** to illustrate the pancellular versus partial distributions of microtubule remnants characteristic of control versus IDA erythrocytes. Red: β-tubulin; green: filamentous (F) actin. Graph: % erythrocytes with pancellular microtubule remnants. Numbers indicate mean ± SEM; $n = 3$ independent samples per group; ***$P = 0.003$, unpaired two-sided Student's $t$ test. See also Supplementary Figs. 2–4. Source data are provided as a Source data file.

disintegration of the microtubule cytoskeleton, but did induce its regrowth over the ensuing 48 h (Fig. 2a). Specialized tissue processing and fixation requirements for microtubule visualization precluded confirmatory studies on patient marrow samples. However, prior studies have shown that freshly obtained human peripheral red blood cells (RBC) retain remnant microtubule fragments that are disrupted by the microtubule destabilizer nocodazole[30–33]. IF for β-tubulin on RBC from non-anemic control patients confirmed the previously described pancellular stippling in the majority (>80%) of cells. Notably, similarly processed RBC from IDA patients lacked these structures in the majority (>80%) of cells (Fig. 2b, c), providing evidence for in vivo erythroid microtubule perturbation with iron restriction.

We then addressed whether erythroid microtubule disruption by iron restriction could impair transferrin receptor trafficking, potentially hindering erythropoietic rescue by holotransferrin. Flow-based quantitative cell imaging, using Amnis ImageStream® analysis, enabled high-throughput and precise measurement of transferrin receptor (CD71) localization in erythroid progenitors. These studies showed that iron restriction ± isocitrate treatment did not affect receptor localization with regard to surface versus intracellular distribution or with regard to colocalization with lysosomes (Supplementary Figs. 3 and 4). As expected, iron restriction did induce a pancellular CD71 increase, which was reversed by the isocitrate treatment (Supplementary Fig. 4). These findings are consistent with the studies of Schuh demonstrating that transferrin receptor vesicular transport employs microtubule-independent, actin-mediated mechanisms[34].

**Identification of FTH1 as a microtubule stability factor controlled by iron and isocitrate.** Iron restriction caused no identifiable alterations in erythroid tubulin levels, isoform composition, or posttranslational modifications; notably, TUBB2A, an erythroid tubulin regulated by GATA1 and heme[35], increased with differentiation but lacked responsiveness to decreasing TSAT (Supplementary Fig. 5a, b). Interrogation of proteomic datasets from staged human erythroid progenitors[18] for annotated MAPs revealed predominance of a destabilizing factor Stathmin (STMN1), with negligible expression of stabilizing MAPs (Fig. 3a). Expansion of the search to include noncanonical MAPs identified Ferritin heavy chain (FTH1), known to possess microtubule binding and bundling activity[20,36–39], as equivalent in abundance to STMN1 at most stages (Fig. 3a). Proteomic data mined from lineage comparisons among human marrow progenitors[19] revealed that the pattern of elevated Stathmin and FTH1 with diminished canonical MAPs was more pronounced in erythroid as opposed to non-erythroid lineages (Supplementary Table 1). In our erythroid culture system, FTH1 demonstrated down-modulation by iron restriction, evident within the first 18 h, and rescue by isocitrate treatment (Fig. 3b and see Supplementary Fig. 5c for antibody specificity). Iron restriction also promoted the appearance of a slightly smaller FTH1 species (arrow in Fig. 3b), that results from NCOA4-mediated lysosomal delivery[40–42]. Isocitrate treatment did not block formation of this fragment despite its complete rescue of the full-length form by day 2. By contrast, a lysosomal protease inhibitor CA074-me prevented formation of the fragment but only partially rescued full-length FTH1, consistent with combined proteolytic and translational control by iron restriction (Fig. 3c). Ferritin light chain (FTL) also displayed modulation by iron restriction and isocitrate treatment (Supplementary Fig. 5d). To assess the consequences of FTH1 loss on microtubules, progenitors underwent lentiviral short hairpin-mediated knockdown (LV-shKD). Effective knockdown with two different hairpins in erythroid progenitors in iron-replete medium caused extensive microtubule disruption with significantly

diminished microtubule density per cell (Fig. 3d, e), supporting FTH1 as a stabilizing element.

To confirm prior reports of ferritin colocalization with microtubules[38,39], we conducted IF co-staining on human progenitors cultured 2–3 days in erythroid medium ± iron restriction and isocitrate treatment. Because of technical limitations with anti-FTH1 antibodies (inconsistent patterns based on source and lot), FTL was used as a marker for ferritin complexes. These studies demonstrated that a subfraction of the cellular ferritin pools clearly colocalize with microtubules (Supplementary Figs. 6 and 7). In cells subjected to iron restriction and isocitrate treatment, the colocalization could be observed in centriolar regions and in nascent microtubules (Supplementary Fig. 6).

**FTH1 levels influence erythroid proliferation and differentiation.** Key phenotypic features of the erythroid iron restriction response include proliferation and differentiation defects that are reversible with isocitrate treatment[11]. A role for FTH1 deficiency in these defects was examined using LV-shKD. Loss of FTH1 in iron-replete progenitors did not significantly affect viability, but diminished proliferation by ~2-fold (Fig. 4a, b). In addition, FTH1 knockdown impaired erythroid differentiation, as reflected by glycophorin A (GPA) acquisition (Fig. 4c, d). Conversely, to determine the consequences of overexpression on the iron restriction response, progenitors underwent transduction with lentiviral ferritin expression vectors lacking upstream iron regulatory elements[22]. Enforced expression of FTH1 alone failed to restore differentiation in iron-restricted erythroblasts; FTL alone similarly lacked rescue activity (Supplementary Fig. 8a, b). However, co-expression of FTH1 and FTL significantly enhanced erythroid differentiation in iron-restricted, but not in iron replete, progenitors (Fig. 4e–g and see also Supplementary Fig. 8c, d).

**Complementary targeting of FTH1 by fumarate and isocitrate enables cooperative reversal of the iron restriction response.** Prior studies have shown that the metabolite fumarate induces FTH1 expression through multiple mechanisms that complement those of isocitrate[11,21,22], raising the possibility for synergistic targeting of the erythroid iron restriction response. Accordingly, iron-restricted erythroid progenitors received varying doses of isocitrate, alone or with fumarate, followed by assessment of FTH1 expression. By this approach, we identified doses of isocitrate (3 mM) and fumarate (1 mM) that separately exerted no significant effects, but together completely restored FTH1 levels (Fig. 5a, b). FTL expression responded in a similar manner (Supplementary Fig. 9a). The doses identified approximate endogenous metabolite concentrations measured in erythroid progenitors. Specifically, we have published isocitrate levels in iron-replete cells to be ~0.2 nmol/$10^6$ cells (~1–2 mM), with a significant, threefold decrease with iron restriction[11]. Fumarate levels in iron-replete cells were found to be ~0.4 nmol/$10^6$ cells (~2–3 mM), with an insignificant decrease associated with iron deprivation (Supplementary Fig. 9b).

Cooperation between isocitrate and fumarate was also evident in the microtubule response to iron restriction, which was reversed by combination treatment, but not significantly affected by either compound alone (Fig. 6a). Similarly, in the rescue of erythroid differentiation during iron restriction, fumarate potentiated the activity of isocitrate (Supplementary Fig. 9c, d). One of the pathways that might contribute to this cooperation could involve fumarate inhibition of Keap1, which mediates cytoplasmic retention, as well as ubiquitination and degradation of an *FTH1* transactivator Nrf2 (refs. [22,43]). However, immunoblots on whole cell lysates showed no evidence of Nrf2 stabilization with

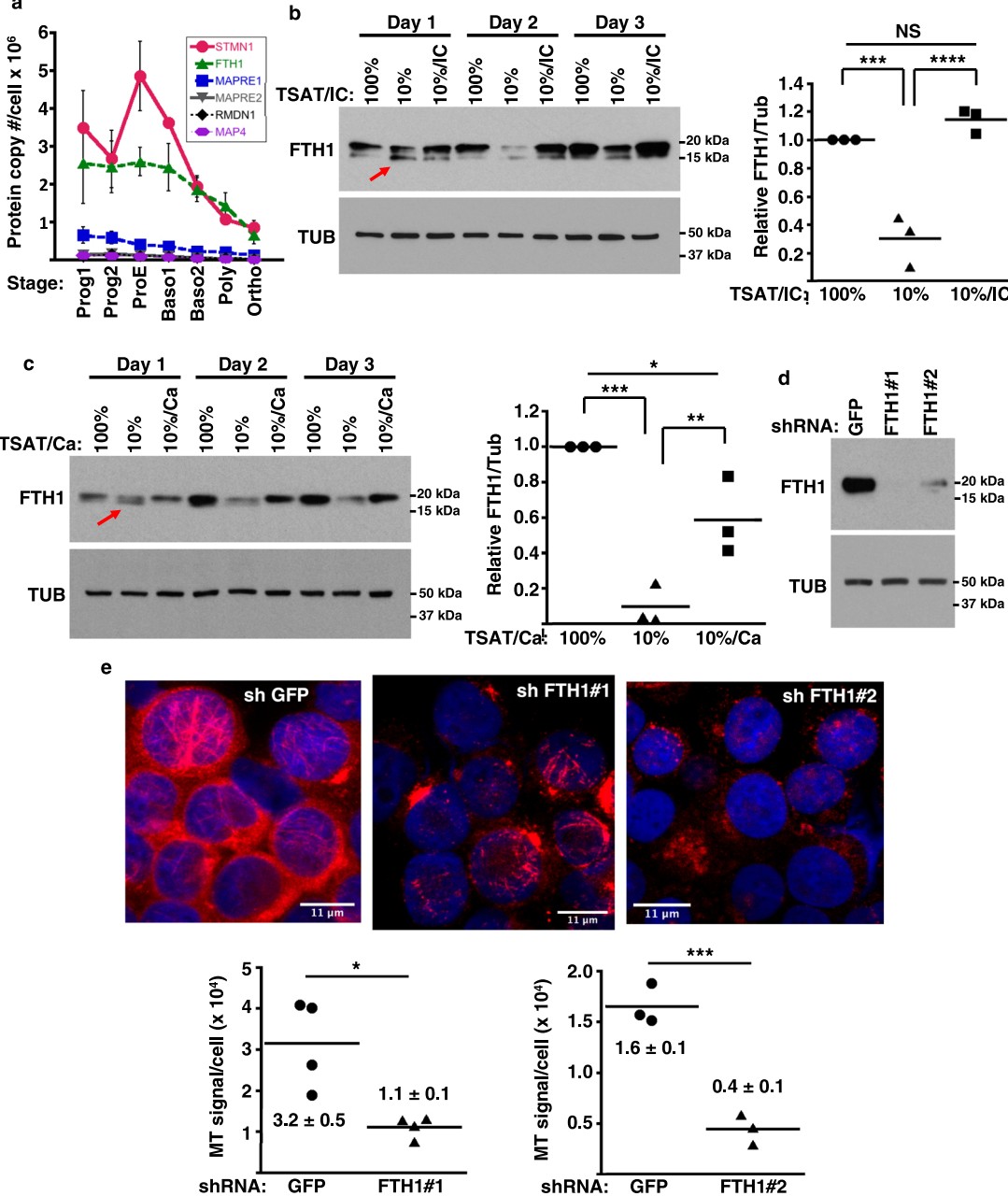

**Fig. 3 FTH1 as a microtubule-stabilizing factor controlled by iron and isocitrate. a** Levels of major microtubule-associated proteins (MAPs) in erythroid cells cultured from human CD34+ progenitors, from proteomic data of Gautier[18]. Graph: mean protein copy number per cell for five most abundant MAPs at indicated stages of differentiation. Red: Stathmin 1 (STMN1); green: ferritin heavy chain (FTH1); blue: microtubule-associated protein RP/EB family member 1 (MAPRE1); grey: microtubule-associated protein RP/EB family member 2 (MAPRE2); black: regulator of microtubule dynamics protein 1 (RMDN1); purple: microtubule-associated protein 4 (MAP4). Stages: progenitor 1 (Prog1), progenitor 2 (Prog2), proerythroblast (ProE), basophilic erythroblast 1 (Baso1), basophilic erythroblast 2 (Baso2), polychromatophilic erythroblast (Poly), and orthchromatic erythroblast (Ortho). Error bars, SEM; $n = 3$ biologically independent replicates for Prog1 and Prog2, and four biologically independent replicates for all other stages. **b** FTH1 levels by immunoblot on whole cell lysates from human CD34+ progenitors cultured 1–3 days in iron-replete (100% TSAT) or deficient (10% TSAT) erythroid medium ± isocitrate (IC). Red arrow: lysosomal proteolytic fragment. Graph: mean normalized FTH1 signal (day 2), relative to value with 100% TSAT. Error bars, SEM; $n = 3$ independent experiments; ***, ****$P = 0.0008, 0.0003$, one-way ANOVA with Tukey post hoc. **c** FTH1 levels by immunoblot on human CD34+ progenitors cultured 1–3 days in iron-replete or deficient erythroid medium ± protease inhibitor (5 µM CA074-me). Red arrow: proteolytic fragment. Graph: mean normalized FTH1 signal (day 2), relative to value with 100% TSAT. Error bars, SEM; $n = 3$ independent experiments; *, **, ***$P = 0.028, 0.013, 0.0006$, one-way ANOVA with Tukey post hoc. **d** FTH1 knockdown with lentiviral shRNA. Immunoblot of transduced human CD34+ progenitors cultured 3 days in iron-replete erythroid medium. **e** Microtubule alterations demonstrated by immunofluorescence on cells as in **d**. Red: β-tubulin; blue: DAPI. Graphs: microtubule (MT) density, reflected as averaged microtubule signal per cell in arbitrary units. Numbers indicate mean ± SEM; $n = 4$ independent experiments for assessment of FTH1#1 and three independent experiments for assessment of FTH1#2; *, ***$P = 0.029, 0.0014$, unpaired two-sided Student's $t$ test. Note: most experiments separately assessed FTH1#1 and FTH1#2. See also Supplementary Figs. 5–7 and Supplementary Data Table 1. Source data are provided as a Source data file.

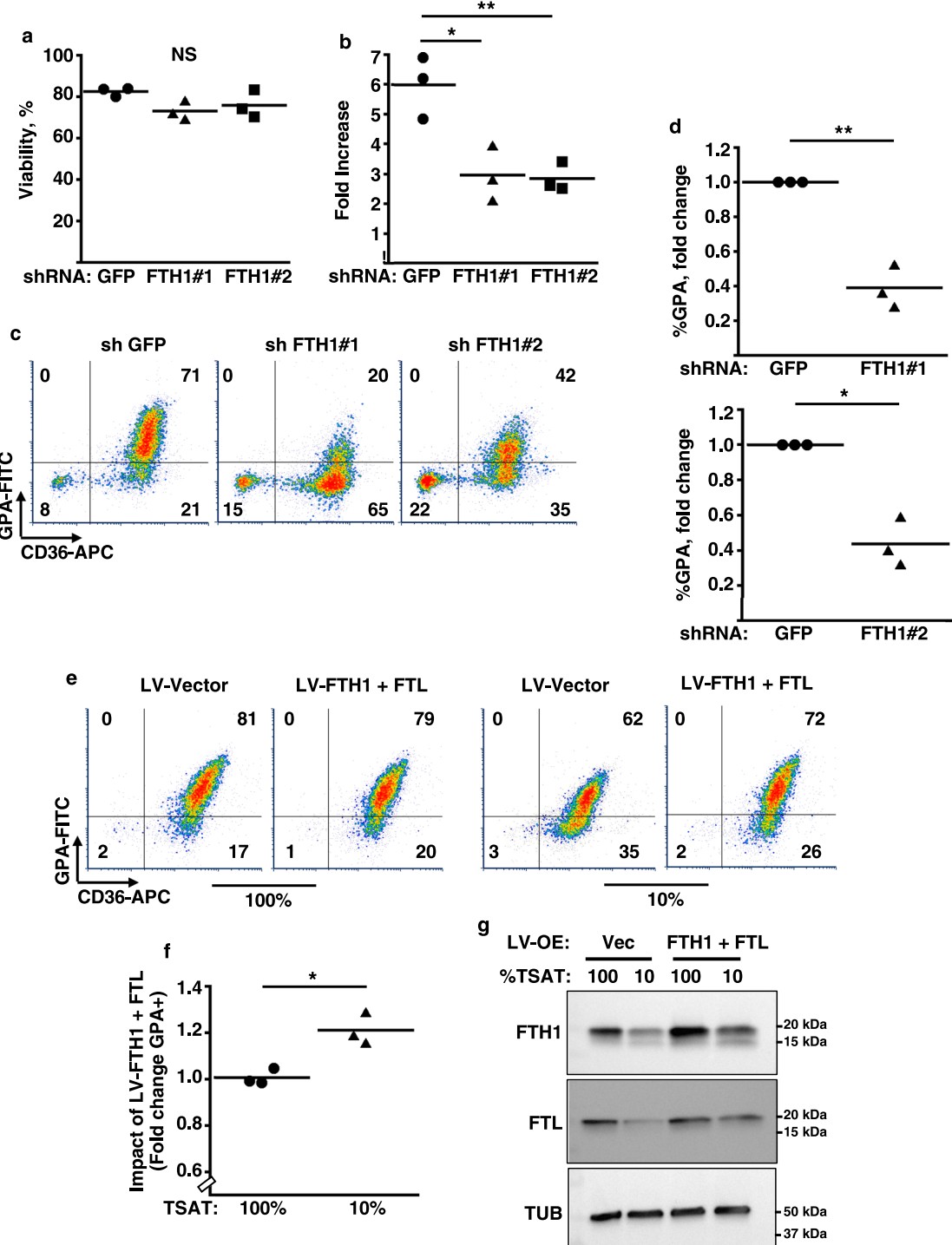

**Fig. 4 Implication of FTH1 in the erythroid iron restriction response. a**, **b** Influence of FTH1 levels on viability and proliferation. Graphs: mean % viable cells and fold increases in cell number for human CD34+ progenitors transduced with lentiviral shRNA vectors targeting either green fluorescent protein (GFP) or ferritin heavy chain (FTH1), and cultured 4 days in iron-replete erythroid medium. Error bars, SEM; $n = 3$ biologically independent experiments; *, **$P = 0.011$, 0.009, one-way ANOVA with Tukey post hoc. NS not significant. GPA glycophorin A. **c** Influence of FTH1 levels on erythroid differentiation as determined by flow cytometry on cells treated as in **a**. **d** Mean fold change in GPA+ cells associated with lentiviral transductions and 4 days culture in iron-replete erythroid medium. Error bars, SEM; $n = 3$ independent experiments; *, **$P = 0.019$, 0.013, unpaired two-sided Student's $t$ test. Note: most experiments separately assessed FTH1#1 and FTH1#2. **e** Distinct consequences of ferritin enforcement in iron-replete versus iron-restricted progenitors. Flow cytometry analysis of CD34+ progenitors transduced with lentiviral (LV) expression vectors for ferritin heavy (FTH1) and light (FTL) chains followed by 4 days culture in iron-replete (100% TSAT) or deficient (10% TSAT) erythroid medium. **f** Mean fold change in GPA+ cells associated with ferritin enforcement in cells cultured as in **e**. Error bars, SEM; $n = 3$ independent experiments; *$P = 0.02$, unpaired two-sided Student's $t$ test. **g** Ferritin levels in cells transduced and cultured as in **e** and **f**, as assessed by immunoblot. LV-OE lentiviral overexpression; Vec vector. Representative results from three independent experiments. See also Supplementary Fig. 8. Source data are provided as a Source data file.

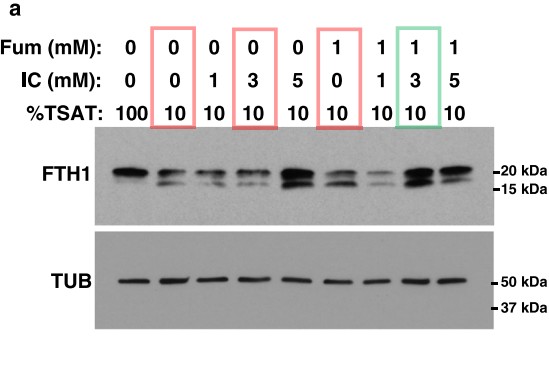

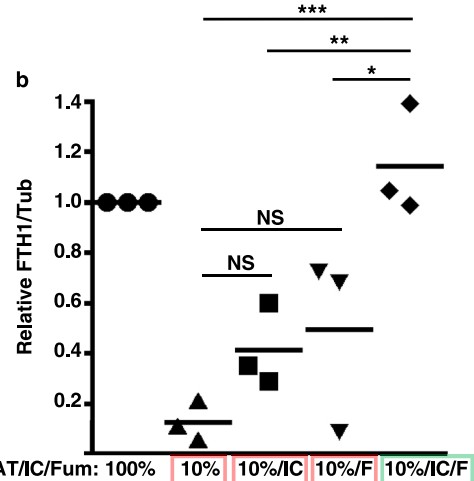

**Fig. 5 Cooperative restoration of FTH1 by isocitrate and fumarate. a** FTH1 levels by immunoblot on whole cell lysates from human CD34+ progenitors cultured 3 days in iron-replete (100% TSAT) or deficient (10% TSAT) erythroid medium ± 1–5 mM isocitrate (IC) and ±1 mM fumarate (Fum or F). Colored boxes highlight synergy of 3 mM IC with 1 mM Fum. Red: iron restriction with no or single metabolite treatment; green: iron restriction with combined metabolite treatment. Representative results from three independent experiments. **b** Graph: mean normalized FTH1 signal (±3 mM IC and ±1 mM Fum), relative to value with 100% TSAT. Error bars, SEM; $n = 3$ biologically independent experiments; *, **, ***$P = 0.02, 0.009, 0.0009$, one-way ANOVA with Tukey post hoc. NS not significant. See also Supplementary Fig. 9. Source data are provided as a Source data file.

fumarate treatment, alone or with isocitrate (Supplementary Fig. 9e); IF showed nuclear predominance of Nrf2 with only minor changes associated with iron restriction and metabolite treatment (Supplementary Fig. 10). Potential alternative mechanisms are discussed below. The potential for in vivo cooperation of isocitrate and fumarate was examined using a murine model of ACDI associated with genetically engineered autoimmune arthritis, i.e., the KRN/G7 strain[44]. Daily intraperitoneal injections of isocitrate alone for 5 days at 150 mg/kg/day had no effect on red cell counts (RBC). However, supplementation of this regimen with minor doses of fumarate (18.6 mg/kg/day) elicited significant RBC increases (Fig. 6b). As will be shown below, fumarate on its own has no effect on RBC counts.

**Fumarate potentiation of isocitrate provides oral efficacy in anemia treatment**. Development of orally active therapies for ACDI has become a major pharmaceutical goal[45]. Accordingly, we tested in murine models the efficacy of isocitrate provided by oral gavage. Gavage administration was not technically feasible

with the KRN/G7 strain due to frequent runting. We therefore adapted prior murine anemia protocols using intraperitoneal administration of killed *Brucella abortus* (KBA)[46,47]. Specifically, maintenance of mice on an iron-restricted diet (48 p.p.m.) combined with weekly low-dose KBA injections ($2.25 \times 10^6$ particles/g) induced a significant, stable anemia (Fig. 7a). The anemic mice responded to isocitrate treatment, but high-dose oral gavage (600 mg/kg/day) provided only transient benefit (Fig. 7a). We therefore employed a regimen of combined low-dose oral isocitrate plus fumarate, following the treatment schedule from the trial in Fig. 6b. This approach yielded a sustained anemia correction, persisting until termination of the trial at 3 weeks after the last dose (Fig. 7b). By comparison, treatment with fumarate alone had no significant effect on the anemia (Fig. 7b). For examination of erythroid progenitor responses, an additional cohort of mice underwent anemia induction and treatment as in Fig. 7b, followed by flow cytometric analysis of marrows and spleens on day 3 after the last dose. Using the maturation scheme of Chen et al.[48], based on CD44 and FSC in the Ter119+ fraction, these studies demonstrated a spleen-specific progenitor response in which the treatment with isocitrate plus fumarate abrogated a block in erythroid differentiation (Fig. 7c, d). Cohorts of mice treated as in Fig. 7b also underwent analysis for treatment effects on the IL-6–hepcidin axis. These studies showed no effect of oral isocitrate plus fumarate on several parameters including: serum IL-6, serum hepcidin, liver hepcidin expression, liver *Tfrc*, or splenic *Tfrc* (Supplementary Fig. 11a–c). These findings, consistent with those of Kim et al. in isocitrate treated mice[14], suggest that the effects are exerted directly on erythroid progenitors. To simulate conditions associated with human chronic anemia management, ACDI mice then underwent extended therapy with oral isocitrate plus fumarate, receiving repeated 5-day treatment courses at a rate of one per month. Strikingly, animals receiving treatment showed complete normalization of RBC counts throughout the 100-day duration of the trial (Fig. 7e).

## Discussion

The erythroid iron restriction response in ACDI results from hepcidin-induced iron sequestration and plays a key role in limiting erythropoiesis[1,3,6,49]. It acts in a lineage-selective manner and desensitizes progenitors to Epo while heightening their responsiveness to inhibitory cytokines, such as IFNγ and TNFα[12,13]. In 66–69% of US and European patients chronically treated with Epo injections for anemia, concomitant IV iron infusions are used to maintain marrow responsiveness, despite the evidence for accumulation of storage iron in tissues[50,51]. Mobilization of this storage iron paradoxically depends on erythroid progenitor Epo responsiveness, which drives ERFE production and consequent hepcidin suppression[3,17,49]. Strategies to break this repressive loop by direct restoration of Epo sensitivity have included treatments with polymeric IgA1, which stimulates EpoR by acting on transferrin receptor 1 (refs. [52,53]), and with isocitrate, which restores surface delivery of EpoR complexes[12,13]. These erythroblast-directed approaches, by preserving pathways of iron homeostasis, offer a potential advantage in a chronic setting over IV iron or recently developed hepcidin antagonists[54], both of which may cause iron overload. Understanding their mechanism of action enables targeted design of Epo sensitizers that will diminish clinical needs for both Epo injections and IV iron, yielding safety and cost benefits.

Epo resistance due to iron restriction results from defects in vesicular trafficking characterized by enhanced lysosomal delivery of Tfr2–Scrib complexes and diminished surface delivery of EpoR[13]. The current studies correlate these trafficking defects

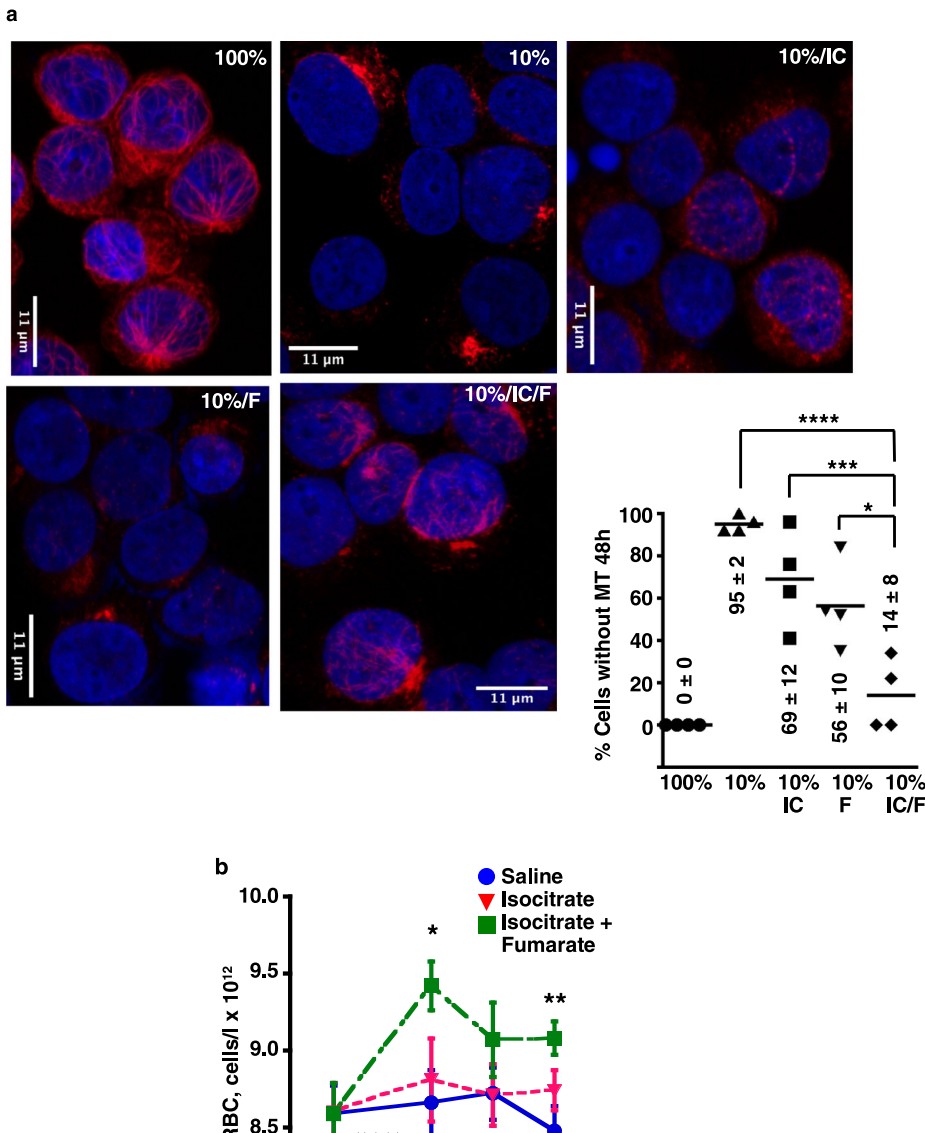

**Fig. 6 Cooperative effects of isocitrate and fumarate on microtubules and on ACDI. a** Microtubule restoration demonstrated by immunofluorescence on human CD34+ progenitors cultured 2 days in iron-replete (100% TSAT) or deficient (10% TSAT) erythroid medium ± 3 mM isocitrate (IC) and 1 mM fumarate (F). Red: β-tubulin; blue: DAPI. Graphs: % cells with undetectable microtubule structures (MT). Numbers indicate mean ± SEM; $n = 3$ biologically independent experiments; *, ***, ****$P = 0.013, 0.0015, 0.0001$, one-way ANOVA with Tukey post hoc test. **b** Amelioration of anemia in a murine genetic model of autoimmune arthritis. Mean red cell count (RBC) in 9-week-old KRN/G7 mice ± isocitrate (150 mg/kg/dose) and fumarate (18.6 mg/kg/dose) provided in five daily intraperitoneal injections (arrows). Blue: saline treatment; red: isocitrate alone; green: isocitrate combined with fumarate Error bars, SEM; $n = 9$/group; *, **$P = 0.04, 0.008$, one-way ANOVA with Dunnett correction using saline control as reference. Isocitrate alone had no significant effect. See also Supplementary Figs. 9 and 10. Source data are provided as a Source data file.

with perturbations in microtubule-dependent structures and implicate FTH1 as a mediator of microtubule responsiveness to iron availability. The prominent role for FTH1 in erythroid microtubule stabilization may arise from an inherent predisposition toward instability in this lineage, as suggested by MAP proteomic profiles (see Fig. 3a and Supplementary Table 1). The mechanism for this function could derive from the capacity of FTH1 to bind and bundle microtubules[20,36,39], in a manner distinct from its classical iron storage function[37]. Supporting such a mechanism, studies have documented iron-independent contributions of FTH1 to internalization of the chemokine receptor CXCR4 (refs. [55–57]) and to cell surface delivery of class I MHC[58];

both these activities depend on microtubule scaffolding[59–61]. However, our results do not rule out microtubule-independent influences of ferritin on EpoR transport, and Golgi vesicle release may occur even with organelle dispersion. Thus, further study is needed to determine relationships among ferritin, microtubules, and EpoR. Independent of effects on EpoR, ferritin also provides a source of iron in heme synthesis through NCOA4-mediated lysosomal trafficking[41,62,63]. Accordingly, erythroid-specific deletion of NCOA4 in mice impairs RBC hemoglobinization[63]. However, erythroid deficiency of NCOA4 also notably enhances Epo sensitivity, as reflected by diminished serum Epo levels with normal RBC and reticulocyte counts[63]. These findings support

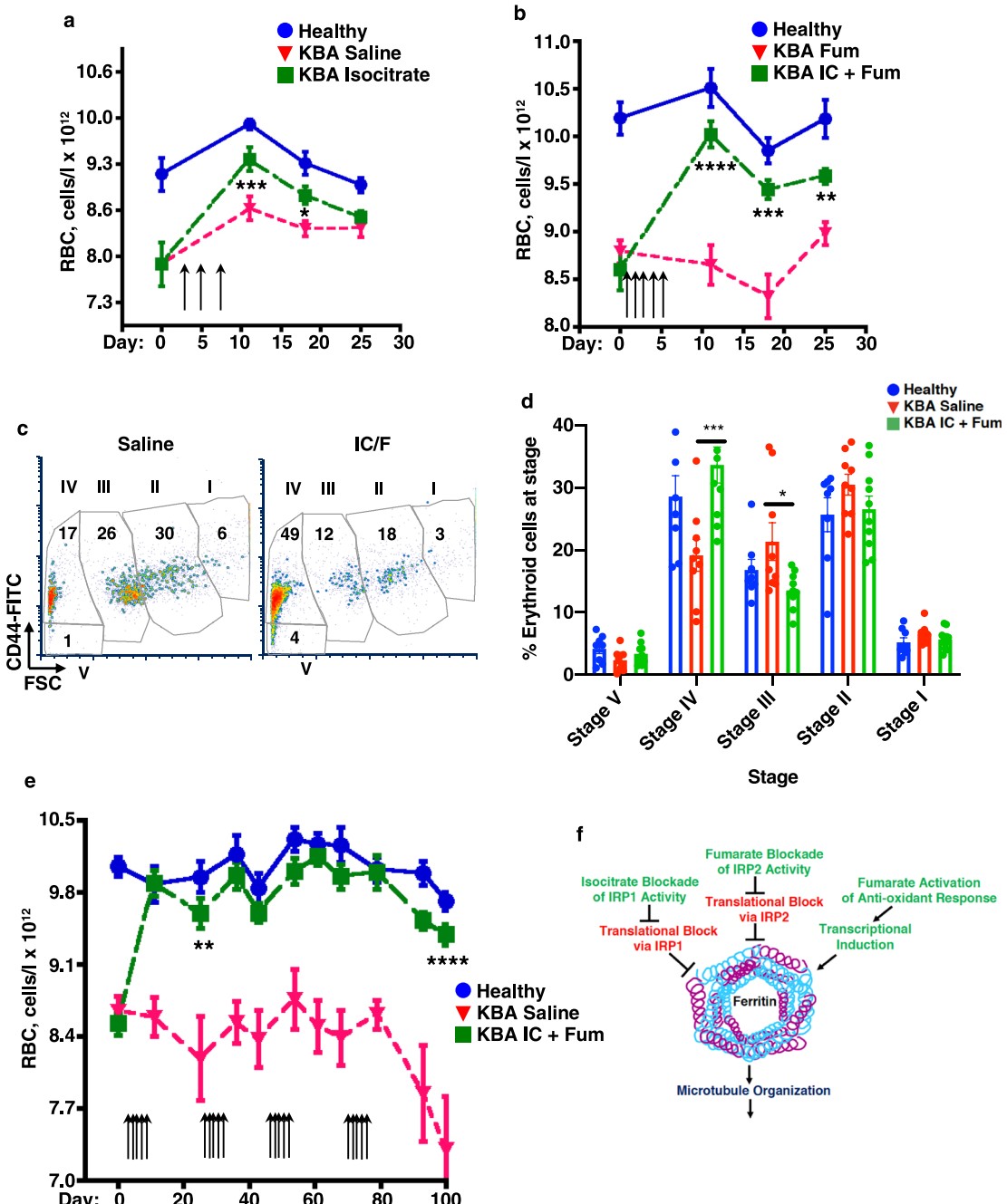

the participation of FTH1 in multiple aspects of erythropoiesis encompassing Epo responsiveness, as well as hemoglobin synthesis.

A major finding of this study concerns the mechanism of action for isocitrate amelioration of iron-restricted anemia, highlighting its abrogation of IRP1 repression of FTH1. We have previously shown isocitrate to selectively inhibit IRP1, but not IRP2 in an *FTH1* IRE RNA-binding assay[11]. Multiple published murine models have supported Irp1 involvement in mediating the erythroid iron restriction response in vivo. Notably, *Irp1*[−/−] mice respond abnormally to iron deprivation, manifesting a paradoxical increase in red cell production[64]. While increased Epo production partially contributes to this phenotype, erythroid progenitor abnormalities are also suggested by the polycythemia seen in 4-week-old *Irp1*[−/−] mice, when serum Epo levels are normal[65]. Conversely, transgenic expression of a constitutively

active Irp1 mutant in mice causes anemia with erythropoietic defects, the basis for which has been unexplained[66]. Several IRP1-independent pathways might also affect erythroid FTH1 levels, including inflammatory mediators, endocrine hormones, and oxidative stress, all of which may influence erythropoiesis[67,68]. Overall, though, the current findings favor a model in which isocitrate correction of iron-restricted anemia occurs largely through reversing IRP1 repression of ferritin, permitting microtubule regeneration (Fig. 7f). Based on this model, synergy with fumarate could arise from targeting of complementary FTH1 regulatory pathways, with fumarate acting on Nrf2 to promote transcription[22,43] and on IRP2 to further enhance translation[22]. Fumarate treatment of erythroid progenitors did not affect overall Nrf2 levels, in contrast to its effects on cultured astrocytes[43]. Under all culture conditions, Nrf2 displayed nuclear predominance, suggesting constitutive release from Keap1. Recently

**Fig. 7 Effective oral anemia therapy with combination of fumarate and isocitrate. a** Transient improvement with oral isocitrate monotherapy. Mean red cell count (RBC) in C57BL/6 mice ± anemia ± isocitrate treatment (600 mg/kg/dose) provided in three gavages every other day (arrows). Blue: healthy non-anemic control mice; red: mice with anemia of chronic disease and inflammation (ACDI) due to injection of killed *Brucella abortus* (KBA), treated with saline; green: mice with KBA ACDI treated with isocitrate. Error bars, SEM; $n = 10$/group for healthy and 9/group for both KBA; *, ***$P = 0.047, 0.0036$, one-way ANOVA with Dunnett correction using KBA saline control as reference. **b** Sustained improvement with oral low-dose fumarate plus isocitrate. Mean red cell count (RBC) in C57BL/6 mice ± KBA ACDI ± fumarate (Fum, 18.6 mg/kg/dose) ± isocitrate (IC, 150 mg/kg/dose) provided in five daily gavages (arrows). Blue: healthy controls; red: mice with KBA ACDI treated with fumarate alone; green: mice with KBA ACDI treated with isocitrate combined with fumarate. Error bars, SEM; $n = 6$/group for healthy and 10–12/group for both KBA; **, ***, ****$P = 0.0036, 0.0003, 0.0001$ one-way ANOVA with Tukey post hoc. **c**, **d** Enhancement of splenic erythropoietic maturation. Flow plots demonstrate maturation stages (I–V) for splenic nucleated Ter119+ erythroblasts from C57BL/6 mice with KBA ACDI treated with saline or isocitrate + fumarate (IC/F) as in **b**. FSC forward light scatter. Graph: mean % erythroblasts at indicated stages for healthy or anemic mice ± isocitrate and fumarate (IC/F). Error bars, SEM; $n = 8$/group for healthy, 9/group for KBA saline, and 10/group for KBA IC + Fum; *, ***$P = 0.026, 0.0039$, one-way ANOVA with Tukey post hoc. **e** Long-term correction of anemia with intermittent oral low-dose fumarate plus isocitrate. Mean RBC in C57BL/6 mice ± KBA ACDI ± fumarate (Fum, 18.6 mg/kg/dose) and isocitrate (IC, 150 mg/kg/dose) provided in five daily gavages (arrows) on the first week of each month. Blue: healthy controls; red: mice with KBA ACDI treated with saline; green: mice with KBA ACDI treated with isocitrate and fumarate. Error bars, SEM; $n = 8$-10/group for healthy; 7–9/group for KBA saline; 8–9/group for KBA IC + Fum; **, ****$P = 0.002$–$0.0001$ for all time points after day 0, one-way ANOVA with Tukey post hoc. **f** Model for potential mechanism of isocitrate–fumarate cooperation in amelioration of iron-restricted anemia. In this model, ferritin serves as a critical node, integrating signals from iron and metabolites and acting on key cytoskeletal functions. Green indicates positive influence, and red indicates negative influence. See also Supplementary Fig. 11. Source data are provided as a Source data file.

published data have revealed transcriptional repression of *FTH1* by Bach1, an Nrf2 antagonist[69], and prior work has shown induction of Bach1 nuclear export by fumarate treatment[70]. Thus, fumarate could potentially derepress *FTH1* through antioxidant pathways targeting the Bach transcription factor family. Ultimately, the synergy between isocitrate and fumarate serves as a prototype for convergent targeting of an iron-sensing cytoskeletal pathway to yield effective, safe, and affordable oral anemia treatments.

## Methods

**Cell culture.** Cryopreserved primary human CD34+ progenitors purified from the peripheral blood of G-CSF-mobilized normal adult donors were purchased from the Fred Hutchinson Cancer Research Center Cooperative Center for Excellence in Hematology (FHCRC-CCEH). For erythroid and granulocytic cultures, these cells underwent incubation for 72 h in pre-stimulation medium consisting of serum-free IMDM (Gibco Thermo Fisher) supplemented with 20% BIT 9500 additive (Stem Cell Technologies), 2 mM L-glutamine (Thermo Fisher Scientific), human SCF at 100 ng/ml, human FLT3 ligand at 100 ng/ml, human TPO at 100 ng/ml, and human IL-3 at 20 ng/ml; all pre-stimulation cytokines were purchased from PeproTech. For the erythroid cultures, the medium was then changed to serum-free IMDM supplemented with 0.05% BSA (Sigma-Aldrich) deionized with Chelex 100 resin (Sigma-Aldrich), 2 mM L-glutamine, ITS additive (Stem Cell Technologies), 0.0012% 1-thioglycerol (Sigma-Aldrich), human Epo at 4.5 U/ml (Procrit), and SCF at 25 ng/ml. For 100% TSAT, the ITS additive consisted solely of ITS-A (Stem Cell Technologies), and for 10% TSAT a 9:1 mixture of ITS-B (Stem Cell Technologies) plus ITS-A was used. Isocitrate treatment entailed inclusion of DL-trisodium isocitric acid (Sigma-Aldrich) at 20 mM, except where otherwise indicated. The fumarate treatment comprised 1 mM sodium fumarate (Sigma-Aldrich). For lysosomal protease inhibition, medium included 5 μM CA074-me (Enzo Life Sciences). For the granulocytic cultures, cells were changed from pre-stimulation medium to serum-free IMDM supplemented with 0.05% BSA deionized with Chelex 100 resin, 2 mM L-glutamine, ITS-A, 0.0012% 1-thioglycerol, human G-CSF (PeproTech) at 10 ng/ml, human SCF at 25 ng/ml, and human IL-3 at 10 ng/ml.

**Immunostaining and microscopy.** Cultured cells resuspended at ~$10^6$/ml underwent cytospin onto glass slides at $1.5 \times 10^5$ cells/slide, followed by 15 min of fixation with 4% paraformaldehyde in PBS (Electron Microscopy Sciences) pre-warmed to 37 °C. After washing three times with PBS, the cells underwent pre-block and permeabilization for 1 h at room temperature in staining buffer (PBS with 2% FBS, 2% BSA, and 0.03% Triton X-100). Staining with primary antibodies diluted in staining buffer occurred at 4 °C for 16 h; phalloidin staining occurred at room temperature for 1 h. The primary antibodies/stains consisted of mouse monoclonal anti-Golgin97 (Invitrogen A-21270) at a 1/100 dilution, rabbit polyclonal anti-Giantin/GOLGB1 (Sigma HPA011008) at 1/100, rabbit monoclonal anti-GM130 (Cell Signaling D6B1) at 1/3000, rabbit polyclonal anti-Pericentrin (Abcam ab84542) at 1/300, rabbit polyclonal anti-CEP135 (Millipore ABE-1857) at 1/100, mouse monoclonal anti-pan-β-tubulin (Sigma T4026) at 1/200, Phalloidin-iFluor 488 CytoPainter (Abcam ab176753) at 1/1000, rabbit polyclonal anti-FTL (Abcam, ab69090) at 1 μg/ml, and rabbit monoclonal anti-Nrf2 (Cell Signaling

Technology, 12721) at 1/1000. After primary staining, slides were washed four times in staining buffer and then incubated with fluorochrome-conjugated secondary antibodies in the dark at room temperature for 1 h. Secondary antibodies included Alexa Fluor® 488-conjugated goat F(ab')₂ anti-rabbit IgG (H + L; Cell Signaling Technology 4412) at 1/300, and Alexa Fluor® 546-conjugated goat anti-mouse IgG (H + L; Invitrogen A-11030) at 1/300. In the final 10 min of incubation, the secondary stain also included DAPI (Sigma) at 1 μg/ml. After washing five times in staining buffer (three times quickly and twice for 5 min each), slides underwent mounting with 15 μl Vectashield® antifade aqueous medium (Vector H-1000) and coverslip. Cells were visualized by confocal microscopy (Ziess LSM-700) using a 63× oil immersion objective, and images were analyzed with Fiji ImageJ version 2.0.0 open-source software (imagej.net/Fiji). For quantitation, ≥50 cells for each experimental condition underwent assessment of reassembled Z-stacks. For Golgi (Golgin97) and centrosomes (Pericentrin), cells were scored negative if no intact organelle could be found. Cells were scored as without microtubules if no tubulin-positive linear structures of any length were found. For comparisons of microtubule density, individual cells were selected and quantitated using Fiji. For superresolution microscopy, slides were imaged on a Zeiss LSM 880 confocal microscope with Airyscan, using a 63× oil immersion objective.

Analysis of preexisting human bone marrow and blood samples obtained for prior clinical indications was carried out in accordance with Institutional Review Board (IRB) approval from University of Virginia (HSR#13310). As discarded and deidentified specimens, they were granted a waiver in this protocol from the informed consent requirement. Adult bone marrow clot specimens from University of Virginia pathology archives were identified by Dr. Alejandro Gru (Division of Anatomic Pathology) based on clinical history, pathology report, slide review and specimen adequacy. ACDI cases consisted of: chronic kidney disease + lymphoma (1), multiple myeloma (1), colon adenocarcinoma (1), rheumatoid arthritis (4), end stage renal disease (2), and empyema (1). For immunohistochemical staining of marrow sections, slides underwent processing on a DAKO Autostainer (Agilent/DAKO), with antigen retrieval at 97 °C for 20 min in high pH TR Flex-DAKO buffer, followed by blocking 10 min in DAKO dual endogenous enzyme block buffer. Primary staining consisted of rabbit polyclonal anti-Giantin/GOLGB1 (Sigma HPA011008) diluted 1/500 in DAKO antibody diluent with incubation for 1 h. Secondary detection employed DAKO Envision anti-rabbit polymer followed by development with DAKO DAB + substrate and hematoxylin counterstain. Freshly obtained (<6 h old), non-refrigerated, EDTA-anticoagulated blood samples from control and IDA subjects were identified and provided by Dr. Doris Haverstick, Director of Clinical Chemistry, University of Virginia Health System. RBC were pelleted by gentle centrifugation at room temperature and resuspended in IMDM. For each slide, $2 \times 10^5$ cells in 650 μl IMDM underwent cytospin and fixation as for the cultured progenitors. Primary staining consisted of mouse anti-pan-β-tubulin as above (1/200 for 16 h), but incubated at room temperature. Secondary staining comprised Alexa Fluor® 546-conjugated goat anti-mouse IgG as above, but also included phalloidin.

**Immunoblot.** Cells washed with ice-cold PBS were resuspended at $10^6$ cells per 100 μl in 1× clear nonreducing Laemmli sample buffer (60 mM Tris-HCl, pH 6.8, 2% SDS, and 10% glycerol) supplemented with PhosSTOP™ phosphatase inhibitors (Roche, 4906845001) and cOmplete™ protease inhibitors (Roche, 11836170001), followed by shearing with tuberculin syringes and protein quantitation with the Pierce BCA assay kit (ThermoFisher, 23225). Prior to loading, samples underwent adjustment with bromophenol blue to 0.01%, dithiothreitol to 100 μM, and boiling

for 5 min. Fractionation of equivalent protein amounts per sample occurred by standard electrophoresis (SDS–PAGE) on 4–15% Mini-PROTEAN® TGX™ precast gels (Bio-Rad) and electrotransfer to nitrocellulose. After washing, membranes were pre-blocked 1 h at room temperature with TBST (20 mM Tris-HCl pH 7.6, 137 mM NaCl, and 0.1% Tween-20) with 5% w/v nonfat dried milk. The primary antibody for FTH1 consisted of a rabbit monoclonal IgG (Cell Signaling Technology, 4393) used at 1/1000 in TBST with 2.5% BSA at 4 °C for 16 h; rabbit polyclonal anti-FTL (Abcam, ab69090) was used under the same conditions. Mouse monoclonal anti-TUBB (Sigma, T4026) was used at 1/250 in TBST with 1% nonfat dried milk at 4 °C for 16 h. Mouse monoclonal anti-TUBB2A (LSBio, LS-C413488) was used at 1/2000 in TBST with 1% nonfat dried milk for 16 h at 4 °C. Rabbit polyclonal anti-AcK379 TUBB2A (LSBio, LS-C412264) was used at 1/500 in TBST with 2.5% BSA for 16 h at 4 °C. Mouse monoclonal anti-TUBA (Sigma, T9026) was used at 1/2000 in TBST with 1% nonfat dried milk for 1 h at room temperature. Anti-AcK40 TUBA consisted of a rabbit monoclonal antibody (Abcam, ab179484) used at 1/1000 in TBST with 2.5% BSA at 4 °C for 16 h. Rabbit polyclonal anti-TUBA, detyrosinated (deY, Millipore, AB3201) was used at 1/500 in TBST with 2.5% BSA for 16 h at 4 °C. Rabbit monoclonal anti-Nrf2 (Cell Signaling Technology, 12721) was used at 1/1000 in TBST with 2.5% BSA at 4 °C for 16 h. Secondary antibodies consisted of HRP-conjugated goat anti-rabbit and anti-mouse IgG (H + L; Invitrogen/ThermoFisher, 31460 and 62-6520) diluted 1/10,000 in TBST with 1% nonfat dried milk and incubated for 1 h at room temperature. After washing of membranes, signal detection employed SuperSignal™ substrates (Thermo, 34580 and 34095). Scanning densitometry with signal normalization for loading and background was conducted using a GS-800 calibrated densitometer (Bio-Rad).

**Lentiviral transduction**. Production of lentiviral particles employed transient co-transfection of HEK293T cells with pCMV-dR8.74 (GAG POL TAT REV), pMD2.G (VSV-G), and vector of interest. Cells grown to 60–70% confluency in DMEM with 10% FBS and 2 mM L-glutamine on 10-cm culture plates received calcium phosphate–DNA complexes assembled with the CalPhos kit (Takara Bio USA, 631312) plus 20 μg plasmid mix (mass ratio of 3/1/4 for pCMV-dR8.74/pMD2.G/vector). Sixteen hours post transfection, medium underwent replacement with Opti-MEM I (Thermo, 31985) followed by 30 h incubation with harvesting of supernatants for filtration (0.45 μ) and storage in aliquots at −80 °C. For transduction, human CD34+ progenitors underwent 48 h culture in pre-stimulation medium, then resuspension in lentiviral supernatants supplemented with pre-stimulation cytokines and addition to six-well culture plates pre-coated with RetroNectin® (Takara, T100A) and loaded with lentiviral particles. Following a 2-h incubation at 37 °C with 5% CO2, the cells received an additional 1 ml of lentiviral supernatant. The medium was then supplemented with protamine sulfate to 5 μg/ml, and plates were spun for 90 min at 1000 × g, followed by overnight culture. The next day, cells underwent resuspension in 1 ml fresh lentiviral supernatant with protamine sulfate, reseeding on the same culture plates, repeat centrifugation, and an additional overnight incubation. The cells, as well as non-transduced controls, were then cultured overnight in pre-stimulation medium with 2 μg/ml puromycin, after which the erythroid cultures were initiated while maintaining selection with 1 μg/ml puromycin.

Lentiviral vectors of interest employed the pLKO backbone. GFP-targeting control shRNA plasmid, pLKO.1 GFP shRNA, was purchased from Addgene (#30323). Human FTH1-targeting shRNA plasmids #1 (TRCN0000029432) and #2 (TRCN0000029433) were purchased from GE Dharmacon (now Horizon Discovery). For FTH1 and FTL overexpression (LV-FTH1 and LV-FTL), the open reading frames of FTH1 and FTL were cloned into pLKO.5-CMV vector[22].

**Flow cytometry**. Cultured human progenitors were washed with PBS and resuspended in ice-cold FACs buffer (PBS with 1% FBS) with fluorochrome-conjugated antibodies added at dilutions of 1/50–1/200 based on prior titrations. After incubation on ice for 30 min, cells underwent washing with FACs buffer and analysis on a FACSCalibur™ flow cytometer (BD Biosciences). Analytical software consisted of FCS Express 6 Flow (De Novo Software), with standard gating and compensation settings[13]. FITC-anti-CD235a (GPA) and fluorochrome-isotype matched control were purchased from ThermoFisher/eBioscience™ (11-9987-82, 11-4732-81). APC-anti-CD36 and matched control were purchased from BD Biosciences (561822, 555585).

Murine spleens for flow analysis were crushed, washed through a 70-μm wire mesh cell strainer with serum-free IMDM, and triturated to achieve a single-cell suspension. To eliminate peripheral blood erythrocytes, cells underwent resuspension in 1 ml room temperature lysis buffer (155 mM NH4Cl, 10 mM KHCO3, 0.1 mM EDTA, pH 7.3) for 3 min, followed by neutralization with 9 ml PBS. Murine marrows were extruded from femurs into serum-free IMDM, with passage through a cell strainer and trituration. Washed cells were stained on ice with antibodies, diluted 1/50–1/200 in FACs buffer, consisting of APC-anti-Ter119 (BD Biosciences 557909), PE-anti-CD71 (BD Pharmingen™ 561937), and FITC-anti-CD44 (BD Pharmingen™ 553133). Matching control rat antibodies were purchased from BD Biosciences. Assessment of maturation within the Ter119+ compartment by CD44 versus FSC employed the strategy of Chen et al.[48].

**Animal model**. All animal experiments were approved by the University of Virginia Institutional Animal Care and Use Committee and comply with all relevant ethical regulations for animal testing and research. The KRN and G7 murine strains, as described[44], were provided by Dr. Kyunghee Choi (Washington University School of Medicine). Male G7 homozygotes were crossed with female KRN homozygotes to generate compound heterozygotes with autoimmune arthritis and ACDI. For induction of ACDI with KBA, we followed the guidelines of Guo et al.[47], with some minor alterations. 7-week-old male and female C57BL/6 mice purchased from Jackson Labs were placed on a 48-p.p.m. iron-restricted diet (Envigo Teklad, Custom TD.80394). After 2 weeks on this diet, we initiated weekly intraperitoneal injections of KBA (Brucellosis Ring Test Antigen, B. abortus strain 1119-3, Lot 1602, USDA, APHIS, NVSL) at a dose of $2.25 \times 10^6$ particles per gram body weight per injection. After 6–8 weeks of injections, with maintenance on the iron-restricted diet, baseline CBC values were obtained. Treatments consisted of sterile saline ± DL-trisodium isocitric acid and sodium fumarate delivered in volumes of 40–200 μl by either intraperitoneal injection or oral gavage. The KBA injections and iron-restricted diet were continued throughout the treatment and posttreatment phases. Retroorbital bleeds into EDTA-coated collection tubes were analyzed for complete blood counts on a VETSCAN HM5 hematology analyzer (Abaxis).

See Supplementary Methods for electron microscopy, imaging flow cytometry, cell metabolite quantitation, and analysis of iron-related parameters in ACDI mice.

**Reporting summary**. Further information on research design is available in the Nature Research Reporting Summary linked to this article.

## Data availability
All relevant data are available from the corresponding author. Source data are provided with this paper.

## Code availability
No codes have been developed for this project.

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

## Acknowledgements

The authors thank Dr. Kyunghee Choi for providing KRN and G7 murine strains; Dr. Doris Haverstick for identifying and providing blood samples; Drs. Shalender Bhasin and

Wen Guo for guidance with the KBA ACDI model; Dr. Stacey Criswell of the UVA Advanced Microscopy Facility for assistance with confocal immunofluorescence microscopy; Dr. Sanford Feldman of the UVA Center for Comparative Medicine for assistance with CBC analysis; Dr. Pat Pramoonjago of the UVA Biorepository and Tissue Research Facility for assistance with immunohistochemical staining of patient marrow samples; Joanne Lannigan and members of the UVA Flow Cytometry Core Facility; Julia Kao for thoughtful discussions; and Dr. Jeffrey M. Macdonald of the University of North Carolina Metabolomic Facility for NMR measurements of isocitrate and fumarate in cellular extracts. This research is supported by the National Institute of Diabetes and Digestive and Kidney Diseases (NIDDK; R01 DK079924 and R01 DK101550), the National Heart, Lung, and Blood Institute (R01 HL149667), National Cancer Institute Center support (P30 CA44579), the Medical Scientist Training Program (5T32GM007267-38), and an NIDDK Research Supplement to Promote Diversity in Health-Related Research (3R01DK079924-09A1S1).

## Author contributions

K.C.F., K.E.E., Z.W.III, and S.K. designed, performed, and analyzed immunofluorescence studies. S.K. performed and analyzed electron microscopy. M.H. conducted imaging flow cytometry studies. K.C.F. and L.L.D. conducted animal experiments. L.L.D. and R.K.S. carried out lentiviral transductions, immunoblots, and flow cytometric assays. A.A.G. and Z.W.III designed and analyzed immunohistochemical staining of patient marrows. M.J.K. and A.O. provided key reagents. A.A. and C.J.L. conducted serum analyte studies in animals. R.P.-G. and N.L. performed transcript studies on livers and spleens from animals. A.N.G. designed and interpreted experiments and wrote paper.

## Competing interests

The authors declare no competing interests.
