## [Peer Review File · Nature Communications]

Reviewers' comments:

Reviewer #1 (Remarks to the Author):

The manuscript by A. Goldfarb and colleagues presents data to support the Golgi disruption and a global collapse of the microtubule cytoskeleton in erythroid iron restriction as it is in the case of anemia of chronic disease and inflammation. They showed that this situation can be reversed by isocitrate and fumarate through the reactivation of ferritin heavy chain (a stabilizing microtubule-associated protein) which is repressed by iron restriction. Based on these results they suggest a potential target for treating iron restricted anemia of chronic disease. The study is interesting and well conducted although the presentation of results is not very clear and fluent.

Major comments

1. Title: it is curious but it does not focus on the real content of the manuscript. I suggest to change it making it more focused. Example: Golgi and microtubule cytoskeleton disruption: new potential targets for treating iron restricted anemia
2. Introduction/results: it appears that most of the work has already been published (introduction) and the results presented in this paper are mainly based on immunofluorescence results, which are interesting but apparently only confirmatory as well as the loss of FTH1. The references reported in the results session are confusing in relation to what is new and what is confirmatory.
3. Discussion: the major outcome of the study, at the end, is that isocitrate ameliorates the iron restricted anemia by abrogating IRP1 repression of FTH1. For the readers this should be the core of the discussion whereas it appears as a sentence at the end after a long discussion which focuses on other aspects. I suggest to reorganize the discussion focusing on the final and most important outcome.

Reviewer #2 (Remarks to the Author):

In this paper, the authors report that iron restriction in erythroid progenitors causes microtubule disassembly and disrupts the Golgi apparatus. The authors provide very interesting data demonstrating that this effect is due to the reduction of ferritin expression, and that the molecular and cellular effects of iron restriction can be reversed by treatment with fumarate and isocitrate that act synergistically on ferritin expression. Importantly, this work provides interesting avenues for oral treatment of anemia and thus is potentially suitable for publication.

However, there are a few conceptual problems that need to be addressed by writing, and some problems that need to be addressed by providing higher quality data and altering the figures.

1. Ferritin has indeed been shown to interact with microtubules in a number of studies, but unfortunately, none of these studies are really convincing. This is most clearly illustrated by the work of Hasan et al *Exp Cell Research* 2006 (ref 42 in the current manuscript). This paper contains some reasonably looking images which quite convincingly show that ferritin does not behave like a proper microtubule-associated protein (MAP) in cells but rather forms blobs, the nature of which is not clear. The specificity of colocalization of ferritin with marginal band microtubules appears more convincing (Infante et al, *Exp Cell Research* 2007), but without clear confirmation of ferritin localisation in the cells studied in the current paper, no strong conclusions about the biochemical activity of ferritin towards microtubules can be drawn. It would be great if the authors could show ferritin localization in their cells in relation to microtubules, even if it doesn't support the conclusion that ferritin is MAP. If this is not possible, the author should change the wording to indicate that the observed effects of manipulating ferritin levels on microtubules may be indirect.
2. Microtubule disassembly and the ensuing Golgi dispersion do not inhibit secretion, especially in small cells – the Golgi stacks re-distribute to the ER exit sites but retain their function. This point should be discussed more clearly. While vesicular transport will certainly be altered after microtubule disassembly, whether this is the cause of alteration in receptor function has not been directly assessed in the paper, and the conclusions and discussion should be moderated accordingly – ferritin might also have more direct effects on some trafficking pathways.

3. High-magnification high-quality immunofluorescence staining images should be shown both for the Golgi and microtubules in all conditions – the quality of the currently shown images is

insufficient for publication.

For example, in Figure 1a, the authors claim that they see Golgi dispersion but this is not visible, especially as the image is dominated by nuclear staining. High-resolution staining of the Golgi channel alone, preferably with at least two Golgi markers, must be included in the main figures. Do the Golgi markers redistribute to ER exit sites, similar to nocodazole treatment? The authors could consider using antibodies against GM130 – the mouse monoclonal antibody from BD Biosciences performs well in isolated cells and in tissues.

The claim of centrosome disruption is not entirely convincing – ninein staining in Extended data to Figure 1 appears reduced but not lost; a centriolar marker, such as CEP135, for which good antibodies are available, should be included in the analysis to investigate whether it is pericentriolar material that is affected, and to which extent. The claim that centrosomes are lost is particularly unconvincing because beta-tubulin staining in what appears to be the centrosome area is still visible in many iron-depleted cells.

Fig 1d: are there any microtubules left in the middle panel? Showing proper high-magnification images is imperative here. It is also unclear what the authors mean by "intact microtubules" in this and other figures – high-quality images accompanied by measurements of microtubule density per cell would be needed.

Extended data 2 – what do the authors mean by "fragmentary" microtubules – better images must be included.

Figure 3e is particularly important to establish causal connection between ferritin expression and microtubules – high resolution images and quantification of microtubule density per cell would be needed here, as there appear to be many dead cells after shRNA-mediated ferritin depletion. Proper scale bars must be included in all cell and tissue images – the currently shown ones are unreadable.

For quantifications where the n number is low (below 20), showing bar plots is unacceptable – dot plots must be shown instead.

Molecular weight markers must be shown for every single Western blot.

4. Writing:

The title should be changed to reflect what is actually shown in the paper - "metabolic-cytoskeletal axis" is too cryptic.

p.3 Scribble and Tfr2 are not "chaperones" in the formal sense, please change.

p. 5 "organelle topology" should be "organelle morphology". Golgi apparatus is not "spherical" although it might appear so at the extremely low magnification currently shown in the paper. Centrosome is not a "Golgi-associated organelle"- Golgi is a "centrosome-associated organelle " in animal cells.

p.13 The statement "Self-assembly of FTH1 into higher order complexes provides multifaceted platforms for assembly of microtubules with one another, with known Kinesin motor partners", should be removed, because although it may sound good, it makes no sense because in cells studied here microtubules do not "assemble with one another with kinesin motor partners" and there is no evidence in the paper that FTH1 forms "multifaceted platforms".

Reviewer #3 (Remarks to the Author):

In this very interesting study Goldfarb and co-workers investigated the association of iron restriction with microtubule composition (Golgi assembly) which could be linked to alterations of Ferritin H expression on a basis on insufficient availability of isocitrate and fumarate. Of note, the authors could also demonstrate that isocitrate and fumarate substitution can be used to treated inflammatory anemia.

Specific points:

Did the defects on microtubule composition upon iron restriction (Fig 1) impact on endocytic

pathways, specifically on uptake of iron loaded transferrin via transferrin receptor and subsequent transferrin receptor trafficking. This would be an important Information for the whole concept of the paper.

In Figure 3 Westernblots for Ferritin H are shown. How can the authors be sure that only FTH is shown and not a mixture of FHL and FTH is presented given that the FTH antibody may cross-react with FTL? Relevant controls are missing pointing to the specificity of their finding. This could be done by showing Western blots for Ferritin H in cells with shRNA knock down also giving an idea on the efficacy of the specific knock down (by RNA and protein quantification for FTH).

The authors present the striking finding that isocitrate and fumarate can efficiently treat inflammatory anemia in mice. In this context important information is lacking which would further strengthen their finding and provide more insights into the mechanisms. I would suggest to show data on the effects of those treatments on hepcidin levels in mice (hepatic mRNA expression), ferritin (FTH/FTL) concentrations in the spleen as a measure for iron restriction or alternatively iron concentrations in organs (spleen /liver) and finally to rule out/ verify that the treatment regimen impacted on inflammation by measuring f.e. Il-6 levels.

Minor:

Page 5. The authors mention that they have studied samples from ACDI patients. No information is given on them in the methods section nor what were the underlying diseases (e.g. cancer, infection, auto-immune..).

Some of the Figures appear to be disrupted as error bars or significance levels are not blotted (f.e diagrams in Figures 1 and 2)

Are the dosages used for isocitrate (3mM) and fumarate (1mM) at a physiological range, what are the concentrations in iron repleted and iron depleted cells?

Reviewer #4 (Remarks to the Author):

This work addresses molecular mechanisms how iron-restriction induces detrimental organellar responses in erythroblasts, and how this mechanism can be modulated or even completely abrogated. The authors identified ferritin heavy chain as a major target of subcellular structure and disintegration. Of note, the addition of isocitrate in addition with fumarate completely prevented the the differentiation block.

Whilst fumarate has originally been introduced into psoriasis on an empirical ground, its pathophysiological activity has been much better understood in other autoinflammatory disease such as multiple sclerosis and its animal model EAE. Here antioxidative response, mediated by the Nrf2 pathway, and shift towards glycolytic metabolic pathways (instead of mitochondrial respiration) have gained increased importance. Since iron metabolism is obviously in the centre of chronic anemia, it may be of interest to characterize the interaction of Nrf2 and Keap in these progenitor cells under iron restriction. This may further help to understand the beneficial activity of this intervention.

The authors wish to thank all of the reviewers for their appreciation of the work and their constructive comments. Wherever relevant, we have addressed these comments with new experimental data. Textual changes, particularly those relevant to the critiques, have been highlighted in blue. Listed below are specific responses to each of the critiques. We believe that this input has led to a greatly improved manuscript and are happy to make any further revisions, as deemed necessary.

Reviewer #1 (Remarks to the Author):

The manuscript by A. Goldfarb and colleagues presents data to support the Golgi disruption and a global collapse of the microtubule cytoskeleton in erythroid iron restriction as it is in the case of anemia of chronic disease and inflammation. They showed that this situation can be reversed by isocitrate and fumarate through the reactivation of ferritin heavy chain (a stabilizing microtubule-associated protein) which is repressed by iron restriction. Based on these results they suggest a potential target for treating iron restricted anemia of chronic disease. The study is interesting and well conducted although the presentation of results is not very clear and fluent.

Major comments

1. Title: it is curious but it does not focus on the real content of the manuscript. I suggest to change it making it more focused. Example: Golgi and microtubule cytoskeleton disruption: new potential targets for treating iron restricted anemia

Reply: we agree and have changed the title to focus on the key points: "Iron Control of the Erythroid Microtubule Cytoskeleton, A Potential Target in Treatment of Iron-Restricted Anemia." See title page.

2. Introduction/results: it appears that most of the work has already been published (introduction) and the results presented in this paper are mainly based on immunofluorescence results, which are interesting but apparently only confirmatory as well as the loss of FTH1. The references reported in the results session are confusing in relation to what is new and what is confirmatory.

Reply: we apologize for confusion over what is published data being cited versus new data being presented. We have modified our terminology in the Introduction (see page 4) and Results (see top of page 5 and middle of page 9) to clarify these distinctions. While immunofluorescence does provide a basis for some of the data, additional approaches include animal model studies, flow cytometric immunophenotyping, immunoblot analysis, EM, and mining of proteomic datasets. In addition, we would assert that immunofluorescence is a very important technique that has played a fundamental role in many cell biology discoveries.

3. Discussion: the major outcome of the study, at the end, is that isocitrate ameliorates the iron restricted anemia by abrogating IRP1 repression of FTH1. For the readers this should be the core of the discussion whereas it appears as a sentence at the end after

a long discussion which focuses on other aspects. I suggest to reorganize the discussion focusing on the final and most important outcome.

Reply: We agree with the importance of this finding and have re-written the Discussion to highlight it. See top of page 16.

Reviewer #2 (Remarks to the Author):

In this paper, the authors report that iron restriction in erythroid progenitors causes microtubule disassembly and disrupts the Golgi apparatus. The authors provide very interesting data demonstrating that this effect is due to the reduction of ferritin expression, and that the molecular and cellular effects of iron restriction can be reversed by treatment with fumarate and isocitrate that act synergistically on ferritin expression. Importantly, this work provides interesting avenues for oral treatment of anemia and thus is potentially suitable for publication.

However, there are a few conceptual problems that need to be addressed by writing, and some problems that need to be addressed by providing higher quality data and altering the figures.

1. Ferritin has indeed been shown to interact with microtubules in a number of studies, but unfortunately, none of these studies are really convincing. This is most clearly illustrated by the work of Hasan et al Exp Cell Research 2006 (ref 42 in the current manuscript). This paper contains some reasonably looking images which quite convincingly show that ferritin does not behave like a proper microtubule-associated protein (MAP) in cells but rather forms blobs, the nature of which is not clear. The specificity of colocalization of ferritin with marginal band microtubules appears more convincing (Infante et al, Exp Cell Research 2007), but without clear confirmation of ferritin localization in the cells studied in the current paper, no strong conclusions about the biochemical activity of ferritin towards microtubules can be drawn. It would be great if the authors could show ferritin localization in their cells in relation to microtubules, even if it doesn't support the conclusion that ferritin is MAP. If this is not possible, the author should change the wording to indicate that the observed effects of manipulating ferritin levels on microtubules may be indirect.

Reply: We strongly agree with the importance of assessing this relationship in our system. Data examining the relative distributions of microtubules and ferritin in our cells under the relevant conditions have been obtained and are in Extended Data Figures 6 and 7. See also text on page 9.

2. Microtubule disassembly and the ensuing Golgi dispersion do not inhibit secretion, especially in small cells – the Golgi stacks re-distribute to the ER exit sites but retain their function. This point should be discussed more clearly. While vesicular transport will certainly be altered after microtubule disassembly, whether this is the cause of alteration in receptor function has not been directly assessed in the paper, and the conclusions and discussion should be moderated accordingly – ferritin might also have more direct effects on some trafficking pathways.

Reply: We greatly appreciate this insight and have moderated our discussion accordingly. Please see the middle of page 15. We have also altered the schematic model in Figure 7f to eliminate the interaction of microtubules with EpoR vesicles, since this interaction remains speculative.

3. High-magnification high-quality immunofluorescence staining images should be shown both for the Golgi and microtubules in all conditions – the quality of the currently shown images is insufficient for publication. For example, in Figure 1a, the authors claim that they see Golgi dispersion but this is not visible, especially as the image is dominated by nuclear staining. High-resolution staining of the Golgi channel alone, preferably with at least two Golgi markers, must be included in the main figures. Do the Golgi markers redistribute to ER exit sites, similar to nocodazole treatment? The authors could consider using antibodies against GM130 – the mouse monoclonal antibody from BD Biosciences performs well in isolated cells and in tissues.

Reply: To improve quality, all immunofluorescent images of progenitors have been replaced by high-resolution Tiffs, depicting the cells at high magnification. Images showing the Golgi channel alone have been included (Extended Data Figure 1a). Additional immunofluorescence experiments have been conducted using GM130 as a Golgi marker (Extended Data Figure 2b, text on page 5). To provide an additional, independent method to assess Golgi status, we have also conducted EM on iron-replete and iron-depleted progenitors (Extended Data Figure 1b, text on page 5).

The claim of centrosome disruption is not entirely convincing – ninein staining in Extended data to Figure 1 appears reduced but not lost; a centriolar marker, such as CEP135, for which good antibodies are available, should be included in the analysis to investigate whether it is pericentriolar material that is affected, and to which extent. The claim that centrosomes are lost is particularly unconvincing because beta-tubulin staining in what appears to be the centrosome area is still visible in many iron-depleted cells.

Reply: We followed this excellent suggestion to analyze CEP135 and demonstrate that the centrioles remain intact with iron deprivation (Extended Data Figure 2c). We have thus amended our description to indicate that iron restriction affects the pericentriolar material (text on page 6). We have eliminated the Figure showing immunofluorescence for ninein due to space constraints and suboptimal quality.

Fig 1d: are there any microtubules left in the middle panel? Showing proper high-magnification images is imperative here. It is also unclear what the authors mean by "intact microtubules" in this and other figures – high-quality images accompanied by measurements of microtubule density per cell would be needed.

Reply: We have provided new high resolution, high magnification images that show residual tubulin-positive aggregates but little to no intact tubular structures (Figure 1d, see text on bottom of page 6). We agree that the term "intact microtubules" is not sufficiently specific. We therefore changed our scoring system to count only cells devoid of any tubular structures, providing for simple, objective and highly reproducible quantitation (see graphs in Figures 1d, 2a, and 6a).

Extended data 2 – what do the authors mean by “fragmentary” microtubules – better images must be included.

Reply: We have eliminated the vague term “fragmentary” and provided a more specific description of the granulocytic microtubules (top of page 7). We have provided higher quality images of the granulocytic microtubules and illustrated their lack of change with iron restriction (Extended Data Figure 2d).

Figure 3e is particularly important to establish causal connection between ferritin expression and microtubules – high resolution images and quantification of microtubule density per cell would be needed here, as there appear to be many dead cells after shRNA-mediated ferritin depletion.

Proper scale bars must be included in all cell and tissue images – the currently shown ones are unreadable.

Reply: We have replaced the photos in Figure 3e to provide high resolution high magnification images, and we have used Fiji to quantitate microtubule density per cell, selecting only viable cells for analysis. We should also note that FTH1 knockdown did not significantly diminish erythroblast viability at day 4 of culture (Figure 4a). We have added appropriate scale bars for all IF images.

For quantifications where the n number is low (below 20), showing bar plots is unacceptable – dot plots must be shown instead.

Reply: All graphs for IF data have been converted to dot plots, with mean and SEM annotated within them.

Molecular weight markers must be shown for every single Western blot.

Reply: Molecular weight markers have been added to all Western blots.

4. Writing:

The title should be changed to reflect what is actually shown in the paper - “metabolic-cytoskeletal axis” is too cryptic.

Reply: Agreed. See reply to Reviewer #1, first comment.

p.3 Scribble and TfR2 are not “chaperones” in the formal sense, please change.

Reply: Chaperone” has been replaced by “associated factors” (page 3).

p. 5 “organelle topology” should be “organelle morphology”. Golgi apparatus is not “spherical” although it might appear so at the extremely low magnification currently shown in the paper. Centrosome is not a “Golgi-associated organelle”- Golgi is a “centrosome-associated organelle ” in animal cells.

Reply: “Topology” has been changed to “morphology” (top of page 5). “Spherical” has been eliminated (top of page 5). The phrase “Golgi-associated organelle” has been eliminated (top of page 6).

p.13 The statement “Self-assembly of FTH1 into higher order complexes provides multifaceted platforms for assembly of microtubules with one another, with known Kinesin motor partners”, should be removed, because although it may sound good, it makes no sense because in cells studied here microtubules do not “assemble with one another with kinesin motor partners” and there is no evidence in the paper that FTH1 forms “multifaceted platforms”.

Reply: This statement has been eliminated, and a sentence has been included to acknowledge that ferritin could exert effects independently of its influence on microtubules (middle of page 15).

Reviewer #3 (Remarks to the Author):

In this very interesting study Goldfarb and co-workers investigated the association of iron restriction with microtubule composition (Golgi assembly) which could be linked to alterations of Ferritin H expression on a basis on insufficient availability of isocitrate and fumarate. Of note, the authors could also demonstrate that isocitrate and fumarate substitution can be used to treated inflammatory anemia.

Specific points:

Did the defects on microtubule composition upon iron restriction (Fig 1) impact on endocytic pathways, specifically on uptake of iron loaded transferrin via transferrin receptor and subsequent transferrin receptor trafficking. This would be an important Information for the whole concept of the paper.

Reply: To address this excellent question, we used imaging flow cytometry (Amnis ImageStream) for high-throughput quantitation of transferrin receptor subcellular distribution in erythroid progenitors under all relevant conditions (Extended Data Figures 3 and 4, text on bottom of page 7 and top of page 8). These results showed that erythroid iron restriction had no effect on transferrin receptor subcellular distribution. This result fits with the finding of Melanie Schuh that the Rab11a-positive vesicles responsible for trafficking the transferrin receptor move by a microtubule-independent, actin-dependent mechanism (Nat. Cell Biol., 2011, 13(12):1431-1436).

In Figure 3 Westernblots for Ferritin H are shown. How can the authors be sure that only FTH is shown and not a mixture of FHL and FTH is presented given that the FTH antibody may cross-react with FTL? Relevant controls are missing pointing to the specificity of their finding. This could be done by showing Western blots for Ferritin H in cells with shRNA knock down also giving an idea on the efficacy of the specific knock down (by RNA and protein quantification for FTH).

Reply: We have added an important control, as requested, showing that shRNA targeting of FTH1 strongly decreases immunoblot signals for FTH1 but has no effect on FTL (Extended Data Figure 5c, text on bottom of page 8 and top of page 9).

The authors present the striking finding that isocitrate and fumarate can efficiently treat inflammatory anemia in mice. In this context important information is lacking which would further strengthen their finding and provide more insights into the mechanisms. I would suggest to show data on the effects of those treatments on hepcidin levels in mice (hepatic mRNA expression), ferritin (FTH/FTL) concentrations in the spleen as a measure for iron restriction or alternatively iron concentrations in organs (spleen /liver) and finally to rule out/ verify that the treatment regimen impacted on inflammation by measuring f.e. Il-6 levels.

Reply: This critical question was addressed by creating a new cohort of mice with ACDI, subjecting them to oral saline versus isocitrate/fumarate treatment, and then analyzing

serum, liver, and spleen for possible changes. These results showed no changes in serum IL-6 or hepcidin with isocitrate/fumarate treatment. Similarly, liver *Hamp* transcript levels did not vary; nor did splenic *Tfrc* transcripts, a marker of the iron pool. Results are shown in Extended Data Figure 10 (text on bottom of page 12 and top of page 13). As indicated, our findings fit with those of Kim et al. (Blood Cells Mol. Dis., 2016, 56:31-36), who showed isocitrate treatment had no effect on circulating iron, storage iron, or inflammation, and did not suppress liver hepcidin mRNA.

Minor:

Page 5. The authors mention that they have studied samples from ACDI patients. No information is given on them in the methods section nor what were the underlying diseases (e.g. cancer, infection, auto-immune..).

Reply: Information has been provided in Methods (text bottom of page 19).

Some of the Figures appear to be disrupted as error bars or significance levels are not blotted (f.e diagrams in Figures 1 and 2)

Reply: Graphs have been revised to provide this information.

Are the dosages used for isocitrate (3mM) and fumarate (1mM) at a physiological range, what are the concentrations in iron repleted and iron depleted cells?

Reply: For isocitrate, we have included a citation from our prior published study (text bottom of page 10). For fumarate, we have provided new data from a metabolomic study (Extended Data Figure 9B and text top of page 11). As indicated, both metabolites are at a physiologic range; endogenous isocitrate, but not fumarate, differs between iron repleted and iron depleted cells.

Reviewer #4 (Remarks to the Author):

This work addresses molecular mechanisms how iron-restriction induces detrimental organellar responses in erythroblasts, and how this mechanism can be modulated or even completely abrogated. The authors identified ferritin heavy chain as a major target of subcellular structure and disintegration. Of note, the addition of isocitrate in addition with fumarate completely prevented the the differentiation block.

Whilst fumarate has originally been introduced into psoriasis on an empirical ground, its pathophysiological activity has been much better understood in other autoinflammatory disease such as multiple sclerosis and its animal model EAE. Here antioxidative response, mediated by the Nrf2 pathway, and shift towards glycolytic metabolic pathways

(instead of mitochondrial respiration) have gained increased importance. Since iron metabolism is obviously in the centre of chronic anemia, it may be of interest to characterize the interaction of Nrf2 and Keap in these progenitor cells under iron restriction. This may further help to understand the beneficial activity of this intervention.

Reply: Because Keap binding destabilizes Nrf2, any change in their interaction should affect Nrf2 levels. Accordingly, we assessed Nrf2 levels in iron-replete versus iron

restricted progenitors and could find no significant difference (Extended Data Figure 9e and text middle of page 11).

REVIEWER COMMENTS

Reviewer #1 (Remarks to the Author):

The authors considered the reviewers comments and they reply in a satisfactory manner

Reviewer #2 (Remarks to the Author):

The authors have addressed most of my comments, but the request to improve microscopy images of the Golgi apparatus were only partially addressed: the immunofluorescence (IF) images of Golgi staining are still of very low resolution, and Golgi dispersion is not visible because the signal is weak. Electron Microscopy images would provide a good solution, but showing two example images is totally insufficient – quantification of multiple EM images would need to be provided. This is obviously a lot of work, so proper IF images would suffice, but high magnification images (preferably as inverted gray scale images) should be shown. Figure 3e – are these arbitrary units? If so, please indicate.

Reviewer #3 (Remarks to the Author):

none

Reviewer #4 (Remarks to the Author):

the authors have provided a substantial revision of their manuscript. With regard to the nrf2 mediated effect this ref. would have wished to see either examination of iron metabolism in nrd2 ko mice or at least confocal microscopy illustrating translocation of nrf2 into the nucleus. A Western Blot on crude cellular material does not differentiate

We would like to thank the reviewers again for their helpful input on our revision. **We have experimentally addressed all of their comments and have included new data and text.** All new text in the main manuscript and the supplemental materials is highlighted in blue. If needed, we are happy to make additional changes and to conduct additional experiments. Our responses below are in blue text.

Reviewer #1 (Remarks to the Author):

The authors considered the reviewers comments and they reply in a satisfactory manner

Reviewer #2 (Remarks to the Author):

The authors have addressed most of my comments, but the request to improve microscopy images of the Golgi apparatus were only partially addressed: the immunofluorescence (IF) images of Golgi staining are still of very low resolution, and Golgi dispersion is not visible because the signal is weak. Electron Microscopy images would provide a good solution, but showing two example images is totally insufficient – quantification of multiple EM images would need to be provided. This is obviously a lot of work, so proper IF images would suffice, but high magnification images (preferably as inverted gray scale images) should be shown.

We have conducted new immunofluorescence experiments and have imaged the cells using superresolution microscopy (Zeiss Airyscan) at the highest magnification, to better illustrate Golgi dispersion with iron restriction. As requested, these images are shown as inverted gray scale, as well as in color. See Extended Data Figure 1a, text in first paragraph of Results (manuscript page 5), and text in Methods (Immunostaining and microscopy).

Figure 3e – are these arbitrary units? If so, please indicate.

We have confirmed in the Legend that the graph in Figure 3e uses arbitrary units.

Reviewer #3 (Remarks to the Author):

none

Reviewer #4 (Remarks to the Author):

the authors have provided a substantial revision of their manuscript. With regard to the nrf2 mediated effect this ref. would have wished to see either examination of iron

metabolism in *nrd2* ko mice or at least confocal microscopy illustrating translocation of nrf2 into the nucleus. A Western Blot on crude cellular material does not differentiate

We have conducted new immunofluorescence experiments and quantified nuclear localization of Nrf2 under all relevant conditions. See Extended Data Figure 10. These results show baseline nuclear predominance of Nrf2 in erythroid progenitors with only minor variations associated with iron restriction and metabolite treatments, as indicated in Results (manuscript page 11). A plausible alternative mechanism of action for fumarate is provided in the Discussion (last paragraph, manuscript page 16). In addition, the diagram in Figure 7f has been revised to remove Nrf2.

REVIEWERS' COMMENTS

Reviewer #2 (Remarks to the Author):

The authors have adequately addressed my concerns and I support publication of this paper.

Reviewer #4 (Remarks to the Author):

the authors have now provided additional analyses of very good quality to show that nuclear translocation of nrf2 is not sufficient to explain the observed synergy of fumarate + isocitrate in this disease model

We reiterate our gratitude to the reviewers for their time, effort, and expertise. Their guidance through this process has really helped to improve the quality of this paper.

REVIEWERS' COMMENTS

Reviewer #2 (Remarks to the Author):

The authors have adequately addressed my concerns and I support publication of this paper.

Reviewer #4 (Remarks to the Author):

the authors have now provided additional analyses of very good quality to show that nuclear translocation of nrf2 is not sufficient to explain the observed synergy of fumarate + isocitrate in this disease model